# Bioinspired engineering of fusogen and targeting moiety equipped nanovesicles

Lixue Wang[1,2,12], Guosheng Wang[2,3,12], Wenjun Mao[2,4,12], Yundi Chen[2], Md. Mofizur Rahman[2], Chuandong Zhu[1,2], Peter M. Prisinzano[2], Bo Kong[5], Jing Wang [6,7] ✉, Luke P. Lee [8,9,10,11] ✉ & Yuan Wan [2] ✉

Cell-derived small extracellular vesicles have been exploited as potent drug vehicles. However, significant challenges hamper their clinical translation, including inefficient cytosolic delivery, poor target-specificity, low yield, and inconsistency in production. Here, we report a bioinspired material, engineered fusogen and targeting moiety co-functionalized cell-derived nanovesicle (CNV) called eFT-CNV, as a drug vehicle. We show that universal eFT-CNVs can be produced by extrusion of genetically modified donor cells with high yield and consistency. We demonstrate that bioinspired eFT-CNVs can efficiently and selectively bind to targets and trigger membrane fusion, fulfilling endo-lysosomal escape and cytosolic drug delivery. We find that, compared to counterparts, eFT-CNVs significantly improve the treatment efficacy of drugs acting on cytosolic targets. We believe that our bioinspired eFT-CNVs will be promising and powerful tools for nanomedicine and precision medicine.

Extracellular vesicles (EV) are cell-derived sub-micrometer-sized vesicles that can mediate intercellular communication by delivering cargo to recipient cells[1,2]. Based on this feature, EVs as potent therapeutic vehicles have been exploited for drug delivery. In contrast with synthetic drug nanocarriers, natural membrane proteins, lipids, and polysaccharides enable EVs to evade phagocytosis, exhibit excellent biocompatibility, own innate stability, and protect encapsulated therapeutics[3–5]. Although EV-based drug delivery is promising, significant challenges are posed by inadequate targeting capability, inefficient cytosolic delivery, low yield, and inconsistencies in preparations, all of which impede the relevant clinical translation[6]. Accordingly, various solutions have been developed to address or alleviate these technical issues. For example, cell-derived nanovesicles

(CNV), as the biomimetics of EVs, have been prepared by mechanical extrusion of donor cells. The generation efficiency of CNVs is enhanced by over 50 to 100 times that of naturally occurring EVs, while the production cost is less than 10% of obtaining natural EVs[7–10]. Our recent study further revealed that CNVs and natural EVs contain over 70% of the same membrane proteins, and the batch-to-batch variation of CNVs can be limited to 10%. These findings demonstrate that CNVs can be excellent substitutes for EVs as drug delivery vehicles. For the mass production of EVs, the stimulation of donor cells and EV-liposome hybrids were also explored[11–16]. Moreover, many techniques have been developed to physically, chemically, and genetically introduce targeting moieties onto EV membranes[17–21], fulfilling active targeting capability. It is noteworthy that EVs preferentially fuse with

[1]Department of Radiotherapy, The Second Hospital of Nanjing, Nanjing University of Chinese Medicine, Nanjing, Jiangsu, China. [2]The Pq Laboratory of BiomeDx/Rx, Department of Biomedical Engineering, Binghamton University, Binghamton, NY, USA. [3]Department of Pulmonary and Critical Care Medicine, Shanghai East Hospital, Tongji University School of Medicine, Shanghai, China. [4]Department of Cardiothoracic Surgery, The Affiliated Wuxi People's Hospital of Nanjing Medical University, Wuxi, Jiangsu, China. [5]Deparment of General, Visceral and Transplantation Surgery, Section of Surgical Research, Heidelberg University Hospital, Heidelberg, Germany. [6]Department of Oncology and Hematology, Yizheng Hospital of Nanjing Drum Tower Hospital Group, Yizheng, Jiangsu, China. [7]Department of Hematology, The Affiliated Drum Tower Hospital of Nanjing University Medical School, Nanjing, Jiangsu, China. [8]Department of Medicine, Brigham and Women's Hospital, Harvard Medical School, Boston, MA, USA. [9]Department of Bioengineering, University of California, Berkeley, Berkeley, CA, USA. [10]Department of Electrical Engineering and Computer Science, University of California at Berkeley, Berkeley, CA, USA. [11]Department of Biophysics, Institute of Quantum Biophysics, Sungkyunkwan University, Suwon, Korea. [12]These authors contributed equally: Lixue Wang, Guosheng Wang, Wenjun Mao. ✉e-mail: dg1535069@smail.nju.edu.cn; lplee@bwh.harvard.edu; ywan@binghamton.edu

homotypic cells[22,23], achieving targeted drug delivery. Nevertheless, the innate homing effect is not comparable to targeting moieties with high binding affinity.

In comparison, limited research focuses directly on EV-based cytosolic drug delivery. Through syncytin, actin, epithelial fusion failure-1, anchor cell fusion failure-1, and other exoplasmic fusogens, EVs can fuse with the recipient cell plasma membrane and directly deliver cargo into the cytosol. But, endocytosis is still the major pathway of EV uptake, and over 70% of endocytosed EVs are re-localized into lysosomes[24–26]. Consequently, the treatment efficacy of vulnerable therapeutics loaded into EVs, such as nucleic acids, peptides, and proteins, is almost inevitably impaired because of drug sequestration and lysosomal breakdown. To optimize the efficiency of EV-cell membrane fusion, viral fusogen expressing EVs, coiled-coil peptide-modified EVs, fusogenic peptide grafted EVs, EV-transfection lipid hybrids, and other derivatives have been constructed[27–31]. These fusogens, lipopeptides, and cationic lipids can interact with complementary proteins or anionic lipids on the recipient cell plasma membranes. The interaction forces the membranes into proximity, which initiates lipid mixing and subsequently pore formation and cargo transfer[32,33]. Yet, these techniques did not adequately address the low EV yield or fulfill active targeting.

Here, we report the bioinspired engineering of fusogen and GPC3-targeted single-chain variable fragment (scFv) co-expressing CNVs, i.e., eFT-CNVs (effective CNVs), for on-target cytosolic drug delivery (Fig. 1). The bioengineered fusogen is a binding-defective but fusion-competent glycoprotein derived from the Sindbis virus[34]. Through the membrane-bound anti-GPC3 scFv, eFT-CNVs can efficiently bind to GPC3 overexpressing cancer cells followed by fusogen-mediated membrane fusion. Meanwhile, the high yield and consistencies of eFT-CNVs prepared by mechanical extrusion can be achieved with low production costs.

## Results

### Cell construction for bioinspired eFT-CNVs
Genetically modified HEK293 cells as donors were constructed (Fig. 2a). Double homozygous knockout (KO) clones were harvested and verified with Sanger sequencing (Fig. 2b and supplementary Fig. 1a). Western blot confirmed that intrinsic GPC3 and B2M were successfully knocked out (Fig. 2c). Subsequently, the double KO HEK293 cells were transfected to express membrane-bound anti-GPC3 scFv followed by fusogen expression. The real-time qPCR results showed that the relative mRNA expressions of anti-GPC3 scFv and fusogen were 237,736 and 1,867 copies. Given that the engineered fusogen contains a 10-residue HA tag, flow cytometry was used to verify further that the HA tag harbored in fusogen was detectable in fusogen and anti-GPC3 scFv co-expressing HEK293 cells, i.e., eFT-HEK293 cells (Fig. 2d). All constructed HEK293 strains were also mycoplasma free (Supplementary Fig. 1b). Next, PKH26-labeled eFT-HEK293 cells or GPC3-targeted scFv expressing HEK293 cells, i.e., eT-HEK293 cells, were seeded into GFP expressing HepG2 cells bound to a petri dish. In a pH 5.5 media, fusogens induced cell-cell fusion by forming multinucleated polykaryons (Fig. 2e). There was no observed cell-cell fusion in the control group that did not contain any fusogen. Numerous studies have verified that a low pH environment is optimal for Sindbis viral fusogen-mediated membrane fusion[35–37]. We are thankful for previous studies on the optimization of the pH for membrane fusion with excellent repeatability. In the other scenario, we thoroughly mixed HepG2 cells and HEK293 or eFT-HEK293 cells in suspension (Supplementary Fig. 2). Without anti-GPC3 scFv present, no cell agglomeration was observed. Moreover, pH did not affect cell agglomeration, indicating anti-GPC3 scFv did not lose its proper folding structure significantly in more acidic conditions. Furthermore, it was shown that HepG2 cells blocked with anti-GPC3 antibodies significantly lose their ability to form microclusters with eFT-HEK293 cells, which was demonstrated by the small prevalence and size of microclusters (ANOVA, $p < 0.05$). In brief, stable eT-HEK293 and eFT-HEK293 cells were successfully constructed, and the membrane-bound fusogen and targeting moiety functioned as expected.

### Preparation and characterization of bioinspired engineering of nanovesicles
Following our well-established protocol[7,8,38], CNVs, eT-CNVs, and eFT-CNVs were prepared by mechanical extrusion of wild-type HEK293, eT-HEK293, and eFT-HEK293 cells (Fig. 3a). Their average sizes were 114.9 nm, 124.3 nm, and 120.9 nm, respectively. It was validated that all vesicles exhibited the characteristic saucer-shaped morphology after observation with electron microscopy (Fig. 3b). The collected CNVs, eT-CNVs, and eFT-CNVs harbor classic EV markers, including membrane-bound protein CD81 and cytosolic protein TSG101 (Fig. 3c). Next, we used anti-HA tag Fab-conjugated gold nanoparticles containing only one antigen-combining region to estimate the fusogen concentration on eFT-CNVs (Fig. 3d). A single eFT-CNV has ~7 fusogen molecules on the plasma membrane ($n = 138$). We further investigated whether eFT-CNVs can fuse to GPC3 expressing HepG2 cell plasma membrane. Approximately $4 \times 10^4$ HepG2 cells were co-cultured with eFT-CNVs in a quantity gradient for 30 min, which is sufficient for GPC3/anti-GPC3 scFv interaction. Signals of HA tag derived from

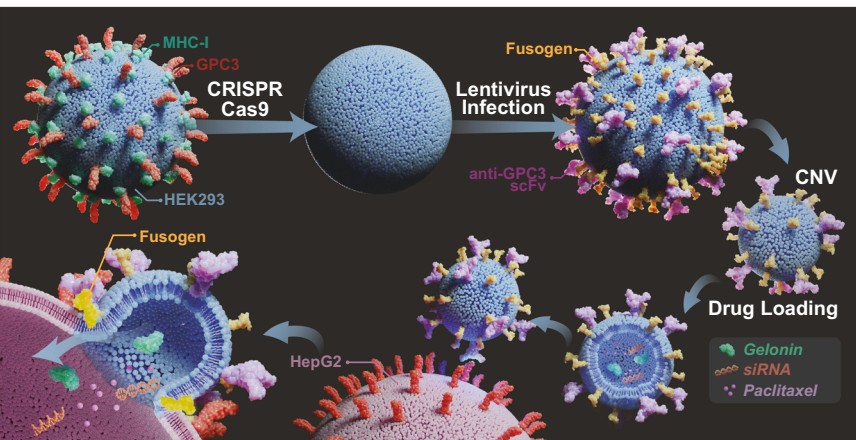

**Fig. 1 | Schematic of anti-GPC3 scoff and engineered fusogen co-expressing eFT-CNVs for cytosolic delivery of therapeutics.** Intrinsic GPC3 and B2M are knocked out using CRISPR/Cas 9 followed by co-expressing anti-GPC3 scFv and engineered fusogen on HEK293 cell membranes. The eFT-CNVs are generated by mechanical extrusion of the donor cells and loaded with various drugs, e.g., nucleic acids, protein toxins, or chemotherapeutic agents, for on-target cytosolic delivery.

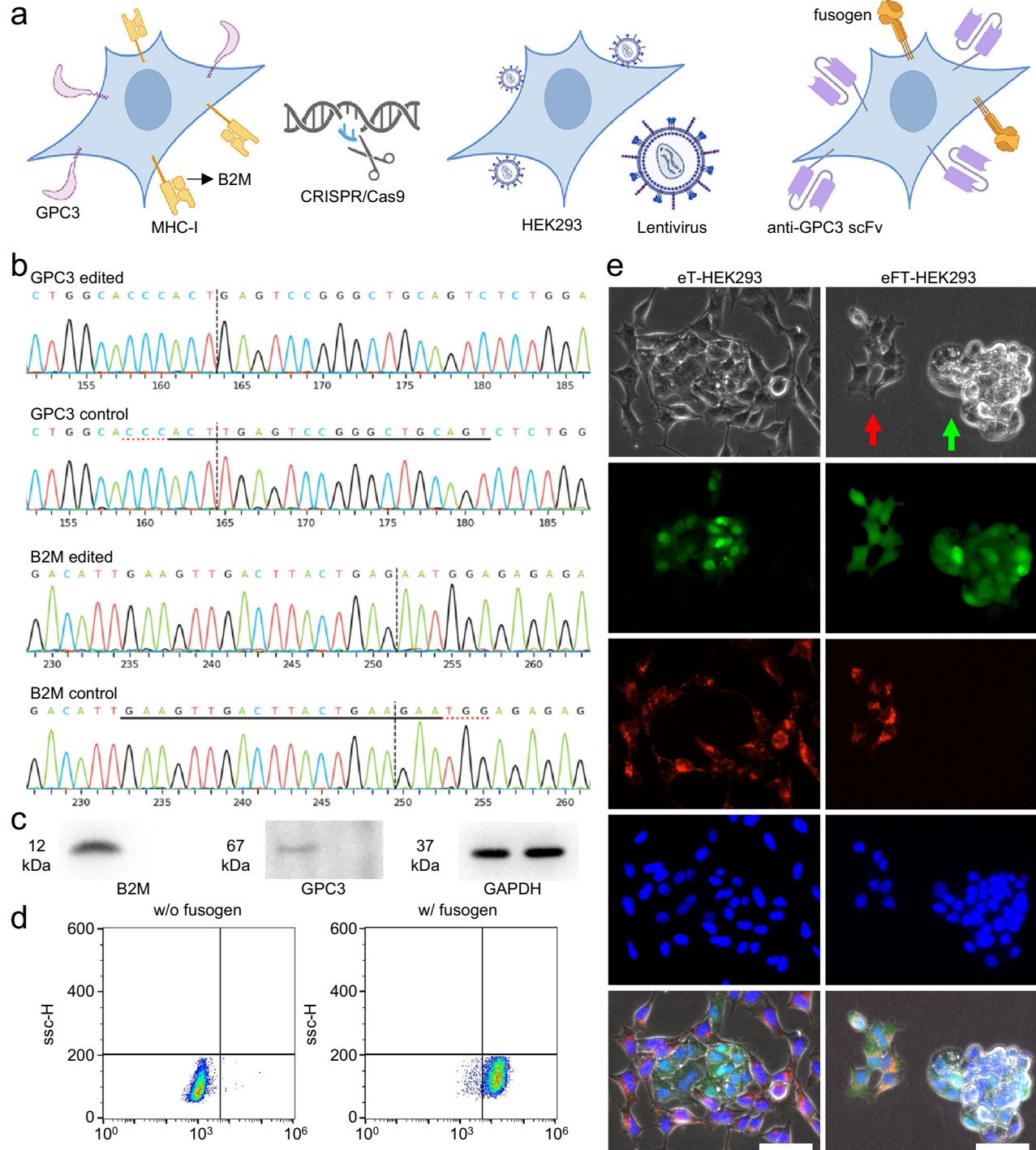

**Fig. 2 | Construction of genetically modified HEK293 cells. a** Schematic of donor cell construction. **b** Sanger sequencing of constructed double KO HEK293 cells. The deleted sequences were underlined. **c** GPC3 and B2M proteins were extracted and identified from wild-type HEK293 cells and double KO HEK293 cells. Experiments were repeated thrice. **d** The expression level of fusogen was measured by flow cytometry in double KO HEK293 cells and eFT-HEK293 cells. **e** pH-dependent fusogen-mediated cell-cell fusion. Cells were co-cultured for 6 h (Green: GFP, Red: PKH26 staining, and Blue: DAPI staining; scale bar is 40 μm). The red arrow indicates multinucleated polykaryons, and the green arrow indicates a HepG2 microcluster. Experiments were repeated five times.

fusogen E2 domain were detected from membrane proteins extracted from treated HepG2 cells (Fig. 3e left). The calculated molecular weight of a HA tag harbored E2 domain is ~49 kDa. The theoretical value was consistent with the experimental result (Supplementary Fig. 3a). The band intensity of the HA tag increased as the amount of eFT-CNVs increased. Based on the signal intensity, semi-quantitative data indicated that ~574, ~126, and ~59 fusogen molecules in each group had

fused to a HepG2 cell. The resulting fusion efficiency was ~22.9%, ~50.4%, and ~100%. The extremely high fusion efficiency in the group of $10^6$ eFT-CNV was attributed to the low amount of eFT-CNVs in only 80 μl of suspension and adequate HepG2 cells in the well. Therefore, almost all eFT-CNVs are efficiently bound to HepG2 cells through anti-GPC3 scFv and fused to plasma membranes. We also cultured HepG2 cells with $1 \times 10^8$ eFT-CNVs over various time intervals. After 20 min

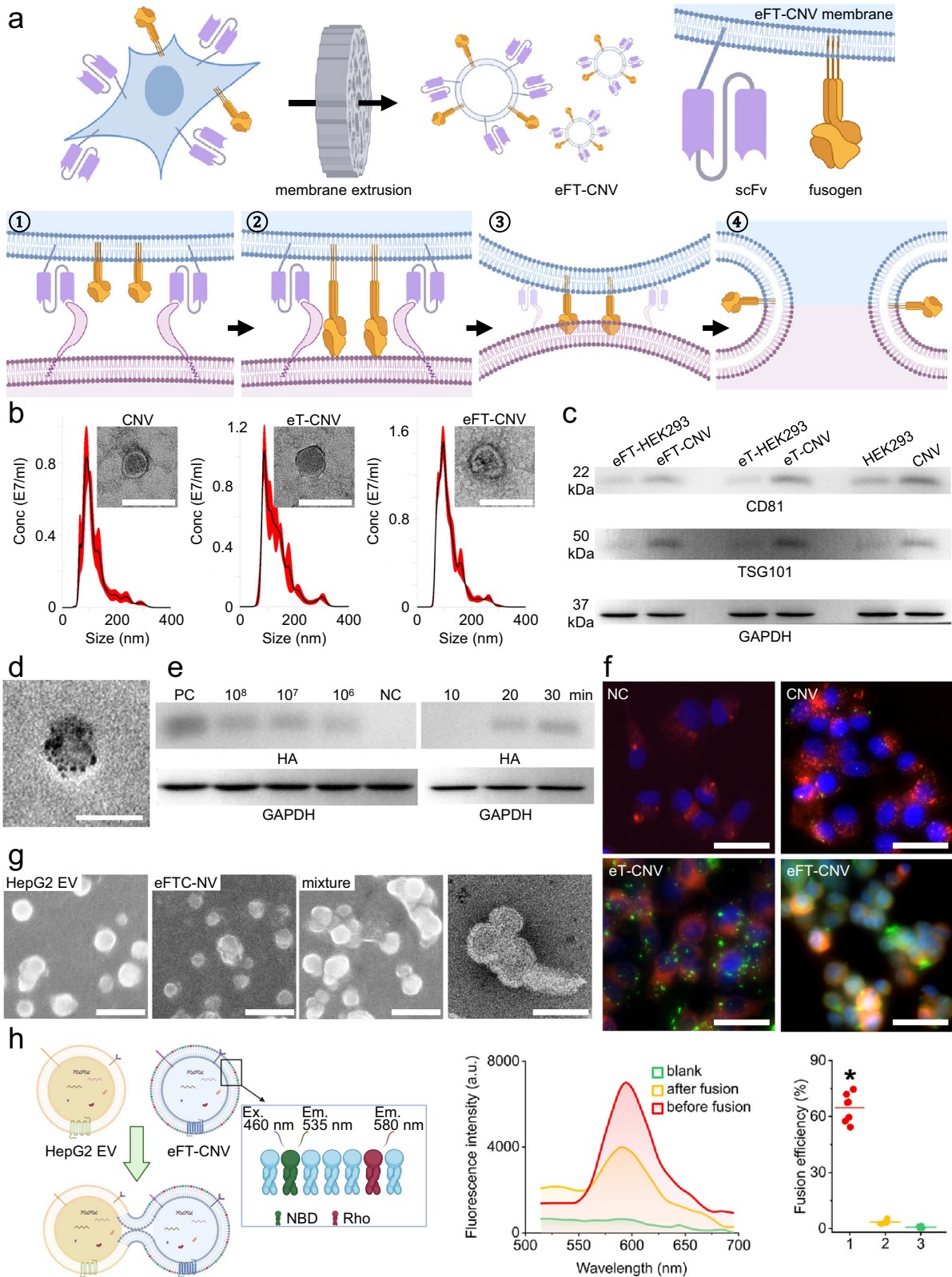

coculture, a sufficient amount of eFT-CNVs fused to HepG2 plasma membranes (Fig. 3e right), exceeding the lowest detection limit of western blot. We speculate that eFT-CNVs can bind to the nearest HepG2 cells in seconds, followed by fusogen-induced membrane fusion. It is noteworthy that the inherent limitations of western blot may not provide accurate data on fusion efficiency between eFT-CNVs and HepG2 cells. Fluorescence imaging techniques also experience

these limitations. Nevertheless, our characterization demonstrated that eFT-CNVs could attach and fuse to the target cell plasma membrane in 20 min. Subsequently, we used a cell assay to verify whether eFT-CNVs can evade lysosomal engulfment. PKH67 labeled CNVs were barely taken up without anti-GPC3 scFv after 30 min coculture (Fig. 3f). Although average 8.3 eT-CNVs quickly attached to HepG2 cell surface, these vesicles without membrane-bound fusogen were endocytosed

**Fig. 3 | Characterization of nanovesicles. a** Schematic of eFT-CNV preparation and mechanism of fusogen-mediated membrane fusion. (1) Anti-GPC3 scFv binds to GPC3 on the target cell; (2) fusogen changes conformation and integrates with target cell membrane; (3) fusogen draws membrane together; and (4) high curvature between the membranes leads to spontaneous fusion. **b** Respective size distribution and morphology of CNVs, eT-CNVs, and eFT-CNVs (scale bar is 50 nm). Experiments were repeated thrice. **c** Western blot analysis of CD81, β-actin, and TSG101 extracted from parental cells and nanovesicles prepared by membrane extrusion. Experiments were repeated thrice. **d** TEM image of eFT-CNVs labeled with anti-HA tag Fab grafted gold nanoparticle (scale bar is 50 nm). Experiments were repeated thrice. **e** Western blot of HA tag and GAPDH extracted from lysates of $1 \times 10^6$-$1 \times 10^8$ eFT-CNVs treated HepG2 cells for 30 min (left). Western blot of HA tag and GAPDH extracted from HepG2 cells treated with $1 \times 10^8$ eFT-CNVs for 10–30 min (right). PC: eFT-HEK293 cell lysate as a positive control; NC: wild-type

HEK293 cell lysate as a negative control. Experiments were repeated five times. **f** $1 \times 10^8$ PKH67 labeled CNVs, eT-CNVs, and eFT-CNVs treated HepG2 cells for 30 min (Green: PKH67 dye staining, Red: Lysoview 594 staining, and Blue: DAPI staining). Experiments were repeated five times. **g** Morphology of HepG2 EVs, eFT-CNVs, and their mixture (scale bar is 200 nm); morphology of fused HepG2 EV and eFT-CNV (scale bar is 100 nm). Experiments were repeated thrice. **h** Fluorescence resonance energy transfer assay monitoring fusogen-induced membrane fusion between HepG2 EVs and lipid dyes labeled eFT-CNVs (scale bar is 40 μm). The fusion changes emission intensity by fluorescent donor NBD and fluorescent acceptor Rhodamine at the excitation wavelength of 460 nm. Plotted curves indicate fusion between dyes labeled eFT-CNVs and unlabeled HepG2 EVs at pH 5.5. The fusion efficiency between HepG2 EVs and (1) eFT-CNVs ($n = 7$), (2) eT-CNVs ($n = 7$), and (3) CNVs ($n = 7$), respectively, at pH 5.5 for 30 min ($p < 0.001$, one-way ANOVA).

by LysoView dye stained HepG2 cells and maintained the form of nanoparticles. On the contrary, only a few eFT-CNVs were endocytosed. Most eFT-CNVs fused to HepG2 cell plasma membranes and presented dispersive green fluorescence throughout the entire cell. The average fluorescence intensity of 200 cells was 22.4. In comparison, CNVs, eT-CNVs, or eFT-CNVs failed to efficiently adhere to GPC3$^{KO}$ HepG2 cell membranes (Supplementary Fig. 3b) after 30 min coculture. We further incubated CNVs, eT-CNVs, and eFT-CNVs with GPC3 expressing MCF7 cells, respectively. Similarly, CNVs barely attached to MCF7 cell membranes; average 3.7 eT-CNVs adhered to MCF-7 cell membranes through anti-GPC3 scFv but did not trigger membrane fusion; and only eFT-CNVs bound and fused to MCF7 cell plasma membranes with average fluorescence intensity of 7.8. It is noteworthy that the average number of eT-CNVs identified in MCF7 cells was significantly lower than that in GPC3 overexpressing HepG2 cells. The average green fluorescence intensity of eFT-CNVs treated MCF7 cells also lower than that in HepG2 cells (Supplementary Fig. 3c). These findings were in line with the differential GPC3 expression level in two cell lines (Supplementary Fig. 3d). The additional analysis of GPC3-mediated endocytosis demonstrated that clathrin-mediated endocytosis, micropinocytosis, and caveolae-mediated endocytosis were involved in the internalization process of eT-CNVs (Supplementary Fig. 3e, f). Lastly, we harvested HepG2-derived GPC3 expressing EVs and mixed them with eFT-CNVs in a 1:1 ratio. Macroscopic agglomeration developed in less than 30 min (Fig. 3g). TEM images further verified the membrane fusion between eFT-CNVs and HepG2 EVs. The fluorescence resonance energy transfer assay confirmed the fusion (Fig. 3h), and the average fusion efficiency at pH 5.5 within 30 min was 64.7%.

**Drug loading and characterization**
Subsequently, we loaded different therapeutics into nanovesicles for cytosolic delivery. Electroporation was used to load siR-Sox2 into CNVs, eT-CNVs, and eFT-CNVs. Meanwhile, sonication was used to load gelonin and paclitaxel. The average loading efficiencies of siR-Sox2, gelonin, and paclitaxel were $3.7 \pm 0.4\%$, $31.2 \pm 2.8\%$, and $27.4 \pm 2.3\%$. Membrane-bound anti-GPC3 scFv and fusogen did not significantly influence loading efficiency. The filtered drug-loaded nanovesicles displayed a saucer-shaped morphology (Supplementary Fig. 4a). The average size of drug-loaded nanovesicles increased, ranging from 8.7 nm to 35 nm (t-test, $p < 0.0001$), but the majority of the size distribution fell in the range of 30–300 nm. After drug loading, the mean zeta potential of drug-loaded nanovesicles also changed accordingly (Supplementary Fig. 4b). Negatively charged siR-Sox2 and paclitaxel further decreased the overall zeta potential ranging from −17.7 mV to −4.4 mV (t-test, $p < 0.05$). In contrast, adding positively charged gelonin slightly increased the overall zeta potential from 1.7 mV to 2.7 mV (t-test, $p < 0.05$). The gelonin and paclitaxel release kinetics from CNVs, eT-CNVs, and eFT-CNVs at 37 °C in pH 5.5 and 7.4 were measured. The fast release rate of gelonin and paclitaxel from nanovesicles

was observed in acidic conditions (Supplementary Fig. 5a). This may be attributed to the relative instability of nanovesicles at low pH. No significant difference in the release rate of gelonin or paclitaxel at 24 h was found among CNVs, eT-CNVs, and eFT-CNVs. Because gelonin is ~30 kDa in size, it cannot freely cross the lipid bilayer compared with small-molecule drugs, so the release rate is relatively slow. At the 24 h time, gelonin-loaded eFT-CNVs released only 10.3% at pH 5.5 and 6.1% at pH 7.4, respectively. The slow release of gelonin from eFT-CNVs is favorable for drug administration. Especially in targeted drug delivery, more gelonin molecules in eFT-CNVs thus can be effectively delivered to the cytosol. All paclitaxel-loaded nanovesicles showed burst release within the first 1 h and then displayed a sustained release profile thereafter. At the 24 h time, paclitaxel loaded eFT-CNVs released ~73.2% at pH 5.5 and 45.3% at pH 7.4, respectively. The siR-Sox2 release profile was not studied due to siRNA degradation.

**Treatment in vitro and in vivo**
First, we verified the treatment efficacy of siR-Sox2 and free gelonin. The CCK8 assay indicated that at the 48 h time, the proliferation of HepG2 cells was inhibited by siR-Sox2 loaded CNVs ranging from $2 \times 10^8$ to $1.28 \times 10^{10}$, demonstrating the anti-cancer effect of siR-Sox2 (Supplementary Fig. 5b). In comparison, an equal amount of mock siRNA-loaded CNVs did not show significant cytotoxicity. We also verified that the free gelonin reduced ribosomal activity (Supplementary Fig. 5c). The IC$_{50}$ of free gelonin was ~16 pM, which is close to the reported value[39,40]. Next, we loaded drugs into $3 \times 10^9$ CNVs, eT-CNVs, and eFT-CNVs, and investigated whether treatment efficacy can be improved by cytosolic delivery (Fig. 4a). The qPCR data showed that siR-Sox2 loaded eFT-CNVs (equivalent to 200 nM siR-Sox2) decreased *Sox2* mRNA levels to 17.6% (Fig. 4b). In contrast, the expression of *Sox2* mRNA in CNV and eT-CNV groups were decreased to 75.7% and 62.6%, respectively. Similarly, Sox2 protein expression decreased to 98% in the CNV group, 76.4% in the eT-CNV group, and 32.8% in the eFT-CNV group. The mRNA and protein expression data demonstrated that siR-Sox2 loaded eFT-CNVs could more efficiently silence the *Sox2* gene in HepG2 cells. We further used 200 nM free paclitaxel, three types of paclitaxel-loaded nanovesicles, and a negative control to treat HepG2 cells. In groups of paclitaxel-loaded CNV, eT-CNV, and eFT-CNVs, 74.5%, 85.1%, and 86.9% of cells were effectively blocked in the G2/M phase. Comparatively, in the free paclitaxel group, only 45.4% of cells were retained in the G2/M phase (ANOVA, $p < 0.001$, Fig. 4c and Supplementary Fig. 5d). Moreover, eT-CNVs and eFT-CNVs can more efficiently deliver paclitaxel to HepG2 in comparison with CNVs (AVOVA, $p < 0.05$). However, no significant difference in treatment efficacy was found between the eT-CNV group and the eFT-CNV group. Given that direct evaluation of the ribosome activity in HepG2 cells is difficult, the protein synthesis inhibition function of gelonin was not directly investigated.

The IC$_{50}$ of free drugs and drugs loaded into three types of nanovesicles was measured. With an equal amount of siR-Sox2, the

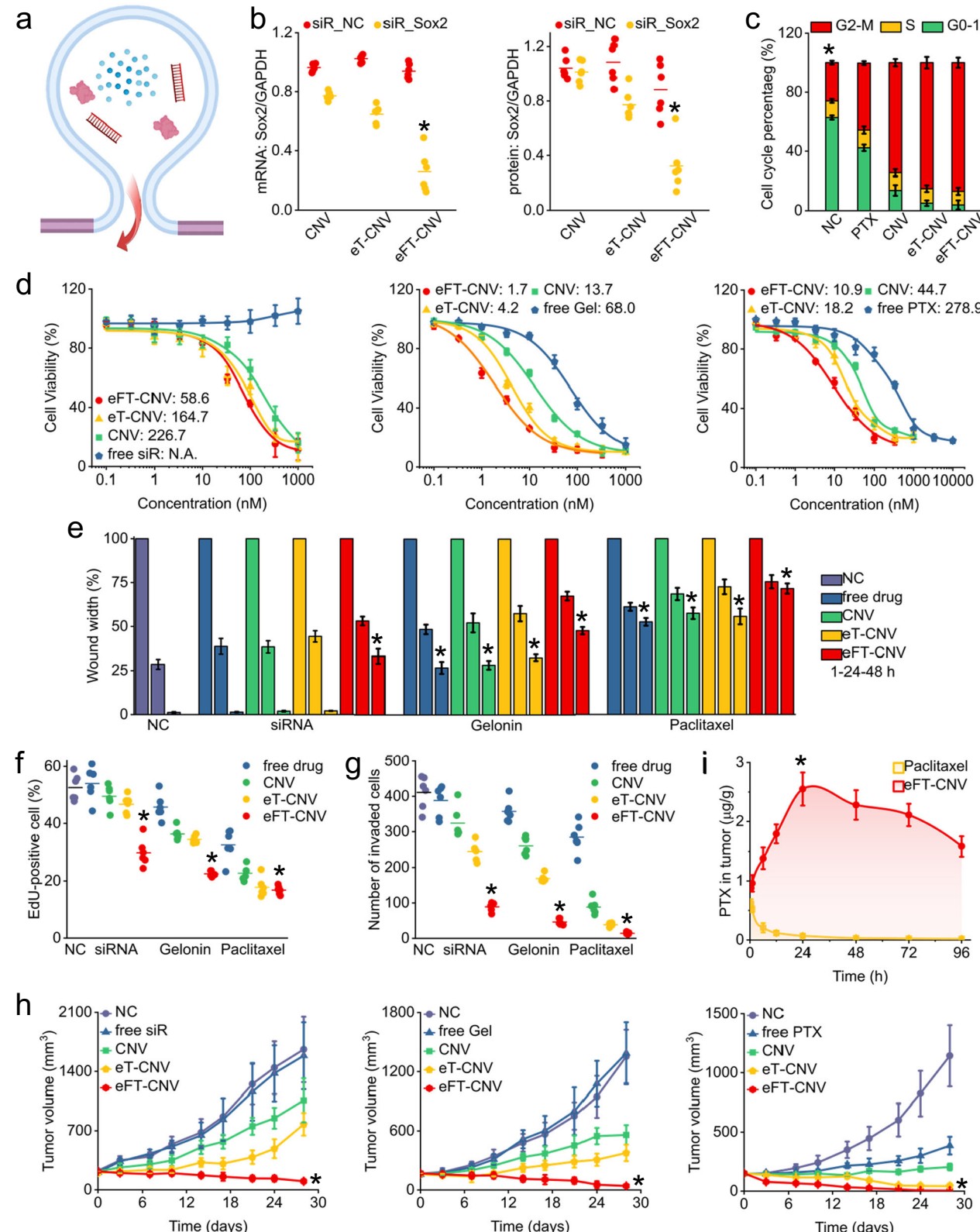

treatment efficacy of drug-loaded eFT-CNVs can be improved by 3.9-fold and 1.4-fold compared to drug-loaded CNVs and eT-CNVs. On the contrary, cell viability was not significantly influenced by free siR-Sox2 (Fig. 4d left). There are 40-fold, 8.1-fold, and 2.5-fold increases in the treatment efficacy of gelonin-loaded eFT-CNVs compared to free gelonin, CNV, and eT-CNV groups (ANOVA, $p < 0.05$, Fig. 4d middle). Similarly, paclitaxel-loaded eFT-CNVs significantly increased

treatment efficacy by 25.6-fold, 4.1-fold, and 1.7-fold compared to free paclitaxel, CNV, and eT-CNV groups (ANOVA, $p < 0.05$, Fig. 4d right). We also performed a wound-healing assay and demonstrated that the wound-closure rate of HepG2 cells decreased after treatment with drug-loaded nanovesicles (Fig. 4e and Supplementary Fig. 6). Particularly, siR-Sox2 loaded eFT-CNVs significantly inhibited HepG2 cell migration by 23.8-fold, 17.5-fold, and 15.9-fold in comparison with free

**Fig. 4 | Characterization of treatment efficacy. a** Schematic of cytosolic drug delivery through membrane fusion for treatment in vitro and in vivo. **b** Gene silencing ability (left) and protein knock-down effect (right) of siR-Sox2 loaded CNVs, eT-CNVs, and eFT-CNVs in HepG2 cells ($n = 6$; $p < 0.05$, one-way ANOVA). The equivalent concentration of siRNA was 200 nM. **c** Quantitative data of cell cycles of $2 \times 10^5$ HepG2 treated with 200 nM free paclitaxel or paclitaxel equivalent in nanovesicles for 48 h, respectively ($n = 3$; $p < 0.05$, one-way ANOVA). **d** IC$_{50}$ of siR-Sox2 (left, $n = 6$), gelonin (middle, $n = 6$), and paclitaxel (right, $n = 6$) loaded three types of nanovesicles treated HepG2 cells, respectively. **e** Quantitative data of wound closure showing free drugs and drugs loaded nanovesicles inhibited migration of HepG2 cells compared to the negative control ($n = 12$; $p < 0.001$, one-

way ANOVA). **f** Quantitative data of cell proliferation of HepG2 treated with drugs and drugs-loaded nanovesicles for 48 h, respectively ($n = 200$; $p < 0.05$, one-way ANOVA). **g** Quantitative data of the invasion assay showing free drugs and drugs loaded nanovesicles inhibited the invasion of HepG2 cells compared to the negative control ($n = 6$; $p < 0.05$, one-way ANOVA). **h** Quantitative data of tumor volume of HepG2 cells xenograft in mice from siR-Sox2 (left, $n = 5$; $p < 0.001$, one-way ANOVA), gelonin (middle, $n = 5$; $p < 0.001$, one-way ANOVA), and paclitaxel (right, $n = 5$; $p < 0.001$, one-way ANOVA) loaded group after drug or placebo administration. **i** Biodistribution of paclitaxel in tumors in free paclitaxel and paclitaxel loaded eFT-CNV groups, respectively, after an intravenous administration ($n = 6$; $p < 0.001$, one-way ANOVA).

siR-Sox2, CNV, and eT-CNV groups (ANOVA, $p < 0.0001$). Gelonin and paclitaxel-loaded eFT-CNVs inhibited HepG2 cell migration by -1.5-fold (ANOVA, $p < 0.05$). An EdU assay verified that HepG2 cells could be more efficiently inhibited by drug-loaded nanovesicles (Fig. 4f and Supplementary Fig. 7a). More specifically, compared to the CNV and eT-CNV groups, the proliferation rate of HepG2 in the eFT-CNV group decreased by 1.7-fold and 1.6-fold in siR-Sox2 groups, 1.6-fold and 1.5-fold in gelonin groups, and 1.4-fold and 1.1-fold in paclitaxel groups (ANOVA, $p < 0.05$). A transwell invasion assay showed that the invasion of HepG2 cells was significantly inhibited by drug-loaded nanovesicles (Fig. 4g and Supplementary Fig. 7b). Compared to CNV and eT-CNV groups, the number of invaded HepG2 in the eFT-CNV group decreased by 3.4-fold and 2.5-fold in siR-Sox2 groups, 3.9-fold and 1.3-fold in gelonin groups, and 5.6-fold and 1.6-fold in paclitaxel groups (ANOVA, $p < 0.05$).

We further investigated the treatment efficacy of drug-loaded nanovesicles in vivo. In siR-Sox2 groups, the tumor volumes of mice treated with PBS and free siR-Sox2 rapidly increased from -250 mm$^3$ on day 0 to -1600 mm$^3$ on day 28 (Fig. 4h left and Supplementary 8a left). Meanwhile, the final tumor volume in CNV, eT-CNV, and eFT-CNV groups was -1060.5 ± 259.3 mm$^3$, 777.1 ± 133.8 mm$^3$, and 101.6 ± 26.4 mm$^3$. A significant difference in tumor volume was found between the eFT-CNV group and the other four groups (ANOVA, $p < 0.01$). The in vivo therapeutic efficacy of siR-Sox2 in the eFT-CNV group was improved 15.7-fold, 10.4-fold, and 7.7-fold compared with the free siR-Sox2, CNV, and eT-CNV groups. Free siR-Sox2 was unable to kill tumors in vivo efficiently. In contrast, siR-Sox2 loaded eFT-CNVs significantly restrained the growth of grafted tumors, which could be attributed to the active targeting effect and cytosolic drug delivery. Similar phenomena were observed in gelonin-treated groups. The final tumor volume in PBS, free gelonin, CNV, eT-CNV, and eFT-CNV groups was 1353.6 ± 271.6 mm$^3$, 1,386 ± 314.2 mm$^3$, 560.8 ± 99.7 mm$^3$, 377.1 ± 83.2 mm$^3$, and 40.6 ± 16.4 mm$^3$ (Fig. 4h middle and Supplementary 8a middle). The in vivo therapeutic efficacy of gelonin-loaded eFT-CNVs was improved 34.1-fold, 13.8-fold, and 9.3-fold compared with free gelonin, CNV, and eT-CNV groups. Tumors treated with free paclitaxel showed a modest growth inhibition; thus, tumor volume increased to 386.7 ± 75.4 mm$^3$. In comparison, paclitaxel-loaded nanovesicles inhibit tumor growth, and the final tumor volume was 205.3 ± 29.7 mm$^3$, 43.6 ± 23.2 mm$^3$, and 4.6 ± 2.7 mm$^3$, respectively (Fig. 4h right and Supplementary 8a right). The in vivo therapeutic efficacy of paclitaxel-loaded eFT-CNVs was improved 84.1-fold, 44.6-fold, and 9.5-fold compared with free paclitaxel, CNV, and eT-CNV groups. No significant difference in body weight among the five groups during the 3-week administration was found. Furthermore, the histologic structures of tumors from all groups were analyzed (Supplementary 8b). Drug-loaded eFT-CNVs caused remarkable tumor tissue damage compared with other groups, indicating enhanced treatment efficacy. Tissue samples were further taken from the major organs of each subject treated with drug-loaded eFT-CNVs for histological analysis. Extensive damage was not observed (Supplementary Fig. 8c), indicating that drug-loaded eFT-CNVs could decrease systemic toxicity. Alternatively, the tumor-bearing mice could receive higher doses

of drug loaded in eFT-CNVs, achieving better treatment efficacy with the systemic toxicity at an acceptable level. There was no significant difference in mouse body weight during the 4-week administration (Supplementary Fig. 9). In addition, paclitaxel-loaded eFT-CNVs continued to accumulate in tumors. They reached peak concentration at a 24 h time point, showing significantly different (t-test, $p < 0.01$) pharmacokinetics compared with the free paclitaxel group (Fig. 4i and Supplementary Fig. 10).

## Discussion

In this study, we constructed a stable HEK293 cell strain that co-expresses anti-GPC3 scFv and fusogen on membranes. Subsequently, we extruded cells and collected the released nanovesicles. Through the membrane-bound targeting moieties and fusogens, drug-loaded eFT-CNVs can efficiently attach to GPC3 overexpressing HepG2 cells, induce membrane fusion, and achieve cytosolic drug delivery. In phase I of cell construction, intrinsic GPC3, and B2M were knocked out. HEK293 cells express a small amount of GPC3 on membranes to maintain cellular functions, including cell growth[41]. Overexpressing anti-GPC3 scFv in HEK293 cells would generate nanovesicles presenting both GPC3 and anti-GPC3 scFv on their membranes. The interaction between the two may induce self-agglomeration of nanovesicles, as shown in the Fig. 3g. To avoid this potential issue, we knocked out the intrinsic GPC3 in HEK293 cells. The knockout of MHC-I may further lower the immunogenicity of CNVs. Previous studies have demonstrated that MHC-I knockout enables immune evasion, and MHC-I deficient CAR-T cells and stem cells have been developed accordingly[42,43]. The knockout of MHC-I was fulfilled by the knockout of B2M. B2M is an essential subunit of MHC-I molecules which present foreign antigens to CD8$^+$ T cells[43]. The B2M knockout can diminish MHC-I expression without affecting cell self-renewal capacity but lowering the immunogenicity of the allogenic cells. It is noteworthy that the knockout of MHC-I may also lower the immunogenicity of EVs or CNVs. EVs derived from antigen presenting cells and nucleated cells still can present antigens to recipient cells via MHC molecules[44,45]. Given aging cells and apoptotic cells cannot be absolutely excluded from cell culture pools, primarily endogenous antigens, such as abnormal proteins derived from aging cells and cell debris released from apoptotic cells, can be presented by EVs via MHC-I. Subsequently, antigen presenting would further trigger immune responses, including immune clearance. Therefore, MHC-I deficient EVs or CNVs would show lower immunogenicity than counterparts with MHC-I. But we also admit that the change in immunogenicity would be small if massive aging or apoptosis does not exist in donor cells. Regarding the debate of high immunogenicity of CNVs over natural EVs, it might be reasonable. Cell debris generated during the mechanical extrusion and the exposed cytosolic proteins could be presented through MHC-I on CNV membranes and trigger immune response. The flip of inner plasma membrane which may harbor positively charged lipids and integral proteins may also induce immune response. In comparison, naturally secreted EVs have less chance to present debris, cytosolic proteins, or positively charged lipids on membranes if the donor cells are healthy. These speculations further illustrate the need for MHC-I

knockout and purification of CNVs. Notably, the majority of CNVs with positively charged surface can be removed during centrifugation and subsequent filtration.

In phase II, anti-GPC3 scFv and fusogen were separately expressed. The function of anti-GPC3 scFv and fusogen has been widely demonstrated[34,35,46,47], and no further elaboration here. Nevertheless, co-expression through single transfection is challenging but preferred. Moreover, the expression level of fusogen can be optimized assuming that increased expression of fusogen molecules improves the chance of fusion. Furthermore, a recombinant fusion protein containing a cancer-targeting and membrane fusion domain deserves investigation. Sindbis virus fusion protein contains E2 and E1 structural domains[48–53]. The E2 has a cell recognition function, whereas E1 facilitates fusion of the viral and cellular membranes, which requires an acidic environment and a cholesterol membrane component. As a class II fusion protein, Sindbis virus fusion protein has a structural signature of β-sheets forming an elongated ectodomain that refolds to result in a trimer of hairpins. Low pH dissociates E1-E2 heterodimers, triggers E1 homotrimerization and conformational changes, and projects fusion loops in E1 outwards. Subsequently, the fusion loops can be buried in the target cell membrane, initializing membrane fusion. The engineered fusogen used in this study is binding-defective but fusion-competent, which was fulfilled by introducing a HA tag and additional mutations in E2[34,35]. Therefore, anti-GPC3 scFv molecules were co-expressed on membranes to achieve cell recognition function. It is noteworthy that in a parallel study, we constructed a recombinant fusion protein in which anti-GPC3 scFv replaced the whole E2. Compared to the separate expression of anti-GPC3 scFv and fusogen, we expected that the all-in-one fusion protein could more efficiently recognize target cells and induce membrane fusion, making nanovesicles behave like a virus but safer than using a virus. However, this recombinant fusion protein did not show binding or fusion activities. The main reason might be the failure of dissociation between E1 and scFv. Leveraging advanced biotechnology, we assume the proposed fusion protein with full function could be constructed. Lastly, not all diseases provide an acidic condition like a tumor microenvironment, which allows pH-dependent fusogen to work efficiently. Therefore, to widely apply this drug delivery system, the pH-independent fusogens are highly desired.

Fusogen or fusogenic peptide-expressing EVs, coiled-coil peptide-modified EVs, and EV-transfection lipids hybrids have been developed for cytosolic delivery. Towards clinical translation, these existing drug delivery systems must address a few inherent issues. The existing fusogen expressing EVs, such as vesicular stomatitis virus G protein (VSV-G), a class III fusion protein, decorated EVs, only reach the tumor lesion through enhanced permeability and retention effect, a passive targeting strategy. Meanwhile, VSV-G EVs arbitrarily attach to both cancer and noncancerous cells through low density lipoprotein receptor which is expressed in almost all tissues, and thus lead to poor specificity of drug delivery[27]. Although coiled-coil peptide-modified EVs demonstrated excellent membrane fusion through the interaction of membrane-bound lipophilic peptides[54], the complementary peptides, must be separately grafted to EV and target cell plasma membrane first, making this approach impractical. EV-transfection lipids hybrids were also developed. Typically, EVs were fused with transfection lipids, such as DOTAP[11]. Through interaction between the target cell plasma membrane and transfection lipids, the encapsulated drugs can be delivered to the cytosol. Yet, the membrane fusion efficiency was less than 30% after 2 h incubation[11,12,55]. In brief, the existing engineered EVs did not fully demonstrate the capability of cancer-targeting cytosolic drug delivery. The membrane fusion efficiency was also modest. These shortcomings, on the contrary, reflect the strengths of our eFT-CNVs. One more point to emphasize is that the mechanical extrusion of donor cells can generate a large number of nanovesicles in only a few minutes, and the

average batch-to-batch variation can be well controlled with optimal manufacturing parameters.

Our study demonstrated that drug-loaded nanovesicles achieved enhanced treatment efficacy compared to free medications. Moreover, compared to nanovesicles without targeting moieties or fusogens, eFT-CNVs can further improve treatment efficacy by delivering drugs to cytosol directly. The effect was particularly obvious in the delivery of gelonin. Gelonin lacks carbohydrate-binding domains, and thus it cannot penetrate cell plasma membranes, making it ineffective in cell treatments. Although free gelonin can be endocytosed, this protein toxin will be delivered into an endosomal compartment and degraded there before it can reach cytosolic targets. Therefore, the ability to evade endocytic vesicles is crucial. The constructed eFT-CNVs exactly fix this gap. Gelonin exhibited potent cytotoxicity when it was directly delivered to the cytosol. The treatment effectiveness of siR-Sox2 was also improved. However, the effect is not as obvious as gelonin's. We speculate that it might be siR-Sox2 itself which does not own high treatment potency. As to paclitaxel, the treatment efficacy of the eFT-CNV group was only increased by 1.7-fold compared to that of the eT-CNV group, indicating that cancer-targeting delivery of small-molecule drugs typically achieves satisfactory effects. Cytosolic delivery can further improve treatment efficacy, but the improvement is modest.

In addition to drug delivery, the developed eFT-CNVs could be used in cell editing and vaccine applications. For example, conferring specific neoantigens to immune cold tumor cell surface could transform them into immune hot ones, encouraging immune cell-mediated tumor killing. The targeted cytosolic delivery of CRISPR/Cas9 complex with eFT-CNVs would enable specific and efficient gene editing. Moreover, compared to lipid nanoparticles, mRNA-loaded eFT-CNVs with low immunogenicity can efficiently deliver mRNA to the cytosol and thus be explored as potential vaccine vectors, e.g., production of CAR-T in situ. Molecular beacons loaded eFT-CNVs can also be used to detect molecules wrapped within EVs. Overall, the fusogen and cancer-targeting moiety co-functionalized nanovesicles as drug vehicles can significantly enhance the therapeutic efficacy of drugs acting on cytosolic targets. We believe our eFT-CNVs could be a valuable and promising tool in nanomedicine and precision medicine.

## Methods
### Generation of B2M/GPC3 knock-out and anti-GPC3 scFv/fusogen knock-in HEK293 cell line
HEK293 cells were ordered from ATCC (CRL-3216). CRISPR/Cas 9 (GenScript, Z03621) was used to construct B2M and GPC3 knockout HEK293 cell strain. The guide sequence 5′-ACU GCA GCC CGG ACU CAA GU-3′ specific to human GPC3 exon 1 and guide sequence 5′-GAA GUU GAC UUA CUG AAG AA-3′ specific to human B2M exon 2 were used. The edited pool was used to seed single cells for clonal expansion. Sanger sequencing was used to verify the cell clones. The sets of PCR primers for GPC3 exon 1 and B2M exon 2 were used as follows: GPC3 F: 5′-CCC TCC CTC AGT AGA CCC AG-3′, GPC3 R: 5′-CAC GTC TCT TGC TCC TCA GG-3′, B2M F: 5′- GGC TTG TTG GGA AGG TGG AA-3′, and B2M R: 5′-CAC GGC AGG CAT ACT CAT CT-3′. Western blot was used to confirm the gene editing effect with HRP labeled antibody (B2M: 1:200, sc-13565, GPC3: 1:200, sc-390587, and GAPDH: 1:500, sc-32233, Santa Cruz). cDNA of fusogen (2961 bp) and anti-GPC3 scFv (1026 bp) were synthesized by GenScript. The constructed pLVX-Hygro plasmid was used (Addgene), and B2M/GPC3 knock-out HEK293 cells were transduced with packaged lentivirus vectors. Subsequently, cells were incubated with 400 μg/ml hygromycin (ThermoFisher, 10687010) for 72 h to enrich hygromycin-resistant cells. The sets of real-time PCR primers for GAPDH, anti-GPC3 scFv, and fusogen were used as follows: actin F: 5′- CCA GCC ATG TAC GTT GCT ATC-3′, actin R: 5′-CTT AAT GTC ACG CAC GAT TTC C-3′, anti-GPC3 scFv F: 5′-GCA CCA GGT TCT ACA GCT AC-3′, anti-GPC3 scFv R: 5′-CAC GGT CAC GCC GAT CAT C-3′,

fusogen F: CCT CTG GCA GCT TTC ATC GT-3', and fusogen R: CAG ATT CAG TGG TGC GTA GC-3'. The constructed HEK293 cells were cultured in DMEM (ATCC 30-2002) supplemented with 0.1% FBS (Thermo-Fisher, 10091148) for 24 h at 37 °C. The cells were labeled with anti-HA antibody (1:200, sc-7392, Santa Cruz) followed by flow cytometry analysis. A PCR method was used to detect mycoplasma in constructed HEK293 cells with F: 5'-TGA AGG TCG GAG TCA ACG GAT-3' and R: 5'-CCT GGA AGA TGG TGA TGG GAT-3'.

## Cell culture and co-incubation

All used cells passed monthly Mycoplasma testing. Cells were maintained in DMEM (ATCC 30−2002) supplemented with 10% (v/v) FBS (ThermoFisher, 10091148), 100 units/ml penicillin (Corning, 30-002-CI), 100 μg/ml streptomycin (Corning, 30-002-CI). Cells were disposed when 50 passages reached. Equal amounts of GFP-expressing HepG2 cells (Angio-Proteomie, cAP-0053GFP) and PKH26 (Sigma-Aldrich, PKH26GL-1KT) labeled HEK293 cells were thoroughly mixed and incubated at 37 °C for 3 h, followed by DAPI (Sigma-Aldrich, D9542) staining for 10 min. Cells were rinsed thrice and seeded onto a poly-d-lysine coated surface for imaging measurements. The geometric mean and median size of cell agglomeration were calculated and plotted. In the other scenario, PKH26 labeled HEK293 cells were seeded in the GFP-overexpressing HepG2 attached petri dish. Cells were co-incubated for 6 h followed by DAPI staining, cell fixation, and imaging.

## Preparation and characterization of HEK293 derived CNVs and HepG2 derived EVs

Three HEK293 cell strains were cultured without FBS till reaching 80% confluence. Harvested cell pellets were resuspended in in-house pre-pared hypotonic buffer containing a proteinase phosphatase inhibitor cocktail (Sigma Aldrich, PPC2020) followed by cell disruption with a Dounce homogenizer (VWR, 71000-514). The supernatant was cen-trifuged at 2,500 g at 4 °C for 15 min followed by 16,500 g for 20 min to discard cellular detritus. Subsequently, the supernatant was filtered using a 0.22-μm pore filter (Sigma-aldrich, SLGVR33RS) followed by ultracentrifugation at 100,000 g at 4 °C for 4 h. The CNV pellets were resuspended with PBS and stored at −80 °C. HepG2 derived EVs were collected following the same procedure. Transmission electron microscopy, Nanosight NS300, and western blot were performed to characterize CNV morphology, size distribution, concentration, and classic EV markers (CD81: 1:500, 52892 S; TSG101: 1:500, 72312 S; Cell Signaling Technology).

## Estimation of fusogen amount on eFT-HEK293 derived CNVs

Anti-HA antibody Fab fragment (H153-70A12, Creative Biolabs) was conjugated to Nanogold in 1.4 nm diameter (2021 S, Nanoprobes) fol-lowed by purification in order to remove unconjugated Fab fragments. Approximately, ~1 × 10^11 Fab grafted Nanogold were incubated with 1 × 10^7 CNVs derived from eFT-HEK293 cells at 4 °C overnight followed by ultrafiltration with Amicon Ultracel-2 centrifugal filter (UFC201024, Millipore). The retrieved Nanogold-labeled CNVs were lyophilized overnight and then imaged under electron microscope. The average Nanogold count per CNV was determined.

## HEK293 derived CNVs uptake by cells and co-incubation with HepG2 derived EVs

Approximately 1 × 10^6, 1 × 10^7, and 1 × 10^8 nanovesicles were incubated with 2 × 10^4 HepG2 cells, GPC3^KO HepG2 cells, and MCF7 cells in a 96-well plate for 10−30 min at 37 °C. Cells were thoroughly rinsed thrice followed by lysis with RIPA (ThermoFisher, 89900). Cell membrane protein was extracted with Mem-PER Plus Membrane Protein Extraction Kit (89842, Thermo Fisher). Protein amount was determined using Micro BCA Protein Assay (23235, Thermo Fisher). Western blot was performed with HRP labeled anti-HA antibodies (1:200, sc-7392, Santa Cruz), anti-GPC3 antibodies (1:500, sc-390587, Santa Cruz), and GAPDH

(1:500, sc-32233, Santa Cruz). Blots were developed with chemilumi-nescence (ThermoFisher, 34094). In parallel groups, ~8 × 10^5 PKH67 labeled nanovesicles were incubated with 2 × 10^4 HepG2 cells, GPC3^KO HepG2 cells, and MCF7 cells in a 96-well plate for 3 h at 37 °C followed by DAPI staining and lysosome staining (LysoView 594, Biotium) for 15 min. After thorough rinsing thrice, fluorescence images were acquired. Green fluorescence dots were counted, and the average intensity of dispersive green fluorescence were measured with ImageJ 1.53t. To investigate interaction between eFT-CNVs and HepG2 EVs, equal amounts of eFT-CNVs and EVs were thoroughly mixed and incu-bated at 37 °C for 6 h. Samples were observed under a scanning electron microscope. In parallel, membrane fusion was monitored by the fluor-escence resonance energy transfer (FRET) assay. eFT-HEK293 cells were labeled with 1% fluorescent lipid NBD-PE (ThermoFisher, N360) and Rho-PE (ThermoFisher, L1392). Through mechanical extrusion, ~1 × 10^8 purified eFT-CNVs were prepared and mixed with equal amounts of HepG2 EVs. The final volume was 200 μl. Fusion efficiency was deter-mined by the change in NBD fluorescence intensity before and after fusion. After fusion, 10% Triton X-100 (Sigma Aldrich, X100) was added to solubilize all nanovesicles. The NBD fluorescence intensity $F_\infty$. The fusion efficiency was calculated by $\frac{F_n - F_0}{F_\infty - F_0} \times 100\%$, where $F_0$ and $F_n$ are the fluorescence intensities before and after fusion.

## Endocytic pathway study

Chemical inhibitors blocking the specific endocytic pathway were used to investigate the endocytic manners of the eT-CNVs. Briefly, 1 × 10^5 HepG2 cells were cultured overnight in a 96-well plate, and pre-incubated with chlorpromazine (Sigma Aldrich, C8138; 10 μg/mL), genistein (Sigma Aldrich, G6649; 1 μg/mL), or cytochalasin D (Sigma Aldrich, C8273; 30 mM). Subsequently, eT-CNVs was added into the plate, and the cells were incubated further for 30 min. Subsequently, cells were stained with DAPI and LysoView for 15 min. Fluorescence images were acquired with Nikon (NIS Elements 5.21.03).

## Drug loading and characterization

siR-Sox2 (5'-CCC GCA UGU ACA ACA UGA UUU-3', IDT), gelonin (ALX-350-150-M001, Enzo Life Sciences), and paclitaxel (S1150, Selleckchem) were used. Drug-loaded nanovesicles were prepared following the widely used protocol[8,56]. The mixture of nanovesicles and siR-Sox2 in a ratio of 10^6 was loaded to a chilled electroporation cuvette followed by electroporation using a BioRad Gene Pluser Xcell system at 1000 kV for 5 ms. After electroporation, the mixture was incubated at 37 °C for 30 min, allowing for the recovery of nanovesicle membrane. Excess siR-Sox2 was removed with a centrifugal filter (50 K MWCO Amicon Ultra-15, Millipore). Gelonin and paclitaxel were loaded with sonica-tion. The mixture was sonicated using a Model 505 Sonic Dis-membrator with a 0.25-inch tip with the following settings: 20% amplitude, 6 cycles of 30 s on/off for 3 min with a 2 min cooling period between each cycle. After sonication, the mixture was incubated at 37 °C for 30 min. The morphology, size distribution, and zeta potential of drug-loaded nanovesicles were characterized. The quantity of loa-ded siR-Sox2, gelonin, and paclitaxel in respective group was mea-sured by Qubit (ThermoFisher), micro-BCA assay, and high-performance liquid chromatography (HPLC, ThermoFisher). To monitor paclitaxel and gelonin release, freshly prepared drug-loaded nanovesicles were placed in a 300 K MWCO float-A-lyzer G2 device (Spectrum Laboratories). The device was then placed in PBS at room temperature with stirring. Samples were retrieved at different time points and analyzed by HPLC or BCA assay.

## Therapeutic efficacy in vitro

Approximately 3 × 10^5 HepG2 cells were treated with ~8 × 10^8 drug-loaded nanovesicles, free drugs, and PBS, respectively. HepG2 cells were treated with siRNA-Sox2 loaded nanovesicles for 48 h at 37 °C. Total RNA was extracted from treated cells with QIAzol (79306,

Qiagen) and measured with Qubit. cDNA was prepared with M-MLV reverse transcriptase kit (M1701, Promega). The primers were as follows: F: 5'- GCT ACA GCA TGA TGC AGG ACC A-3' and R: 5'- TCT GCG AGC TGG TCA TGG AGT T-3'. The relative Sox2 mRNA level was determined by the $2^{-\Delta\Delta CT}$ method. The Sox2 protein level in each group was determined using the ELISA kit (ab245707, Abcam). Evaluation of ribosome-inactivating protein catalytic activity of gelonin was determined by using the cell-free rabbit reticulocyte lysate assay (L4960, Promega). HepG2 cells were treated with paclitaxel loaded nanovesicles (equivalent to 200 nM free paclitaxel) for 48 h at 37 °C. A total of $5 \times 10^5$ fixed HepG2 cells in 100 µl of 1× binding buffer (BD Biosciences, 556454) were incubated with 50 µg/ml PI (BioLegend, 640914) overnight at 4 °C. Cell cycle analysis was performed with flow cytometry (BD Accuri C6). Data was processed with Flowjo 10.4. To determine cytotoxicity of drug loaded nanovesicles, ~1000 HepG2 cells were seeded into each single well in a 96-well plate. Cells were treated with PBS, free drugs, and drug-loaded nanovesicles with a concentration gradient. A CCK8 assay (Dojindo, CK04-01) was used. Cell proliferation was determined with EdU assay (Sigma-Aldrich, BCK-EDU488). To investigate cell migration, 2-well silicon insert (Ibidi, 80209) with a defined 500 µm gap was used. The wound width was monitored under microscope at 1, 24, and 48 h time points and measured with ImageJ 1.53t. A tumor invasion system (Corning, 354167) was used to assess the invasive potential of treated and untreated HepG2 cells. Cells that had invaded through the Matrigel matrix membrane (Corning, 354248) were stained with Wright-Giemsa for 15 min followed by enumeration.

## Therapeutic efficacy in vivo

All relevant ethical regulations for animal testing have been complied with. All animal experiments were approved by and performed by guidelines from the Institutional Animal Care and Use Committee (IACUC) of the Model Animal Research Center of the Second Hospital of Nanjing. The housing conditions for the mice were as follows: 12:12 h dark/light cycle, ambient temperature of $22 \pm 1$ °C, and ~55% of humidity. Approximately $5 \times 10^6$ HepG2 cells in 50 µl of PBS mixed with 50 µl of Matrigel (Corning, 354248) were inoculated subcutaneously to the flanks of BALB/c mice (~18–22 g, 6 weeks, $n = 5$). When tumor volume reached ~100 mm³, mice were randomly divided into 5 groups. The drug was intravenously administrated every 2–3 days (1 mg siRNA-Sox2-, 2.5 mg gelonin-, and 7.5 mg of paclitaxel-equivalent per kg of body weight per dose) for 3 weeks. The tumor volume was calculated by (length × width²)/2. After euthanasia major organs and tumors were collected followed by tissue staining. To investigate the biodistribution of paclitaxel, ~0.2 g of tissue samples were collected at eight-time points after treatment. The amount of paclitaxel in major organs and tumors was measured by HPLC.

## Statistical analysis

All experiments are repeated at least three times. Data was presented as the mean ± standard deviation (SD). Statistical comparisons were performed by two-tailed T-test and one-way ANOVA test. A $p$-value < 0.05 was considered significant.

## Reporting summary

Further information on research design is available in the Nature Portfolio Reporting Summary linked to this article.

## Data availability

All relevant data of this study were presented in the paper and Supplementary Information file. Source data are provided with this paper. Additional information and unique biological materials can be requested from the corresponding author upon request. Further details are described within the Supplementary Information file that includes characterization of cells and nanovesicles, cell assays data, and animal study data. Source data are provided with this paper.

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

## Acknowledgements
Y.W. thanks the support from National Cancer Institute R01CA230339 and R37CA255948. L.W. thanks the support from Jiangsu Provincial Medical Youth Talent (QNRC2016054) and the Leading-Edge Technology Program of Jiangsu Natural Science Foundation (BK20212012).

## Author contributions
L.W., B.K., J.W., L.L., and Y.W. designed the research. L.W., G.W., W.M., Y.C., M.M.R., C.Z., and P.M.P. performed the experiments, collected, and analyzed the data. All authors contributed to the writing of the manuscript, discussed the results and implications, and edited the paper at all stages.

## Competing interests
The authors declare no competing interests.
