## [Peer Review File · Nature Communications]

REVIEWER COMMENTS

Reviewer #1 (Remarks to the Author):

In this study, Wang et. al., reported fusogen and targeting moiety co-functionalized cell-derived nanovesicles (eFT-CNV) for cytosolic delivery of therapeutics. Donor cells were engineered and constructed, which express anti-GPC3 scFv and binding-defective but fusion-competent fusogens. Through mechanical extrusion the derived nanovesicles harboring these biofunctional membrane-bound proteins can be massively produced as nanocarriers. Membrane fusion effect and efficiency were demonstrated by limited but essential evidence. In downstream applications, drug-loaded eFT-CNVs showed improved treatment efficacy, especially for proteins. In addition, eFT-CNVs own the potential to confer proteins to membranes of designated recipient cells, which could temporarily alter cellular phenotype. Therefore, this simple yet sophisticated method in combination with unique properties of extracellular vesicles (EV) can promote EV-centered fundamental and translational research. Overall, this study was well designed with clear logic flow. The chain of custody is relatively intact. A few minor concerns need to be clarified or address.

- 1) To demonstrate the specificity, GPC3 rare expression or null expression cell lines should be used.
- 2) B2M was knocked out. However, it is unclear whether B2M-deficient CNV can significantly promote biocompatibility.
- 3) The author should discuss or show data to indicate whether the extrusion process change the cargos of the CNVs. What type of RNAs and proteins do they carry and are they therapeutic or toxic? Even HEK cells are widely used as an engineered cell line it will be great to check the cargos.
- 4) VSVG has also been proposed as a powerful tool to promote fusion. Please discuss the comparison between VSVG and Fusogen
- 5) It is a clever strategy to use CRISPR technology to remove MHC and GPC3. Can the authors show actually the resulted cells and CNVs are less immunogenic? The field has some debating on the immunogenicity of EVs and CNVs. The current notion is that while allogeneic cells are immunogenic, allogeneic EVs or CNVs are not that immunogenic. It will be crucial if the authors can demonstrate that this removal of MHC and GPC3 are necessary and essential.

Reviewer #2 (Remarks to the Author):

In this research, the authors developed GPC3 (glypican 3)-targetable scFV and Sindbis viral fusogen-expressed cells-derived nanovesicles (eFT-CNVs) for enhancing cytosolic release of nanovesicular contents into the targeted cells. As the authors pointed out, bioinspired vesicles such as extracellular vesicles are highly anticipated to be next-generation intracellular delivery tools, however, low target and cytosolic release efficacy should be improved for attaining the practical usages. The developed technique seems to be useful for further development of vesicle-based intracellular delivery system, however, there are points that should be considered and revised.

(Expression level and linker length) About scFv and fusogen expression, expression level and linker length from plasma membrane might possibly affect the molecular targeting and membrane fusion efficacy. Optimization should be needed.

(Negative control about non-expression of GPC3) There was no data in vitro (cellular uptake and biological activity) and in vivo (solid tumor targeting) about the case of non-expression of GPC3 on cells as negative control. In addition, how about only fusogen expression (without scFV expression) on CNVs in each experiment??

(Supplementary Figure 2) What are “tfHEK293” and “HepG2 GPC3 Blk”?? In addition, detailed data of “only binding on cell surface” and “membrane fusion” should be assessed in each condition.

(Figure 2e and 3f) The experiment for checking cell-cell fusion in only pH5.5 condition. How about pH7 condition?? Negative control experiments should be needed. In addition, how about HEK293 without any expression of scFV and fusogen??

(Figure 4b) Delivery of siR-Sox2 using eFT-CNVs, how about off-target effects??

(Figure 4d and 4e) In Figure 3f, eFT-CNVs showed much higher fusion than that of eT-CNVs. However, in Figure 4d and 4e, effective degree of fusogen expression for biological activity was not so high (ex. PTX: eFT-CNV 10.9, eT-CNV 18.2). Reasons why? Fusogen did not work well in the expression design, or not??

(Figure 4f) Each result of “free drug” in CNC, eT-CNV, and eFT-CNV was very different. Reasons why?

(Supplementary Figure 4) In the experiment of molecular release from each CNV in pH5.5 or pH7.4, pH5.5 condition showed higher release efficacy than that of pH7.4 condition. Reasons why about pH-

dependency? In the case of liposome, pH-dependent release will be also shown, or not?? In addition, retention efficacy of bioactive molecules in CNVs (derived from plasma membranes) and liposome is similar, or not??

(Cellular uptake mechanism) Cellular uptake mechanism of the eFT-CNVs via GPC3 (glypican 3)??

(Supplementary Figure 7) How about concentration dependency of each CNVs about biological anti-tumor activity in vivo?? In addition, in Figure 7C, negative control in each organ should be added. Data of body weight should be also added for assessing side effects as a supplementary figure.

(Supplementary Figure 8) eFT-CMV enhanced retention not only in tumor but also in each organ. Detailed assessment about damage in each organ should be needed.

(Western blot) There were only objective molecular size data in the manuscript. Full molecular size range data should be added as supplementary figures.

(Figure 2d) What are X-axis and Y-axis??

(Supplementary Figure 3) siR-Sox2, gelonin, and paclitaxel were showed to affect zeta-potential. There are high amount of surface binding on vesicular membrane (not encapsulation), or not??

(Supplementary Figure 5 and 6) What are “negative controls” of gelonin and paclitaxel??

Reviewer #3 (Remarks to the Author):

Targeted delivery of small-molecule and large-molecule drugs to specific target cells or organs inside the body is challenging. Delivery vehicles that are derived from fragments of cellular lipid bilayers, such as exosomes or enveloped virus-like particles, are gaining more attention for their ability to selectively deliver molecular cargo to specific target cells. Interestingly, the lipid bilayer can be functionalized with targeting and fusion molecules to enable to programmable delivery of molecular cargo to specific cells based on targeting molecule-ligand interactions.

In the manuscript submitted by Wang et al., the authors present results where cellular membrane fragments, isolated from cells overexpressing targeting and fusogen molecules, are turned into vesicles via extrusion (termed “eFT-CNVs”), thereby scaling particle production over traditional exosomes approaches. The results presented by Wang et al. are interesting, particularly the results from experiments using CNVs to systemically deliver drugs to tumor-bearing mice. The manuscript is well-written and does a good job of citing relevant literature. However, it is difficult to assess the cell-type-specific delivery of eFT-CNVs in the experiments presented by the authors. For example, it would have greatly improved the manuscript if the authors had included an extruded CNV displaying a fusion molecule in the absence of a targeting molecule. That critical control particle would allow for the interpretation of the importance of displaying an antibody-derived targeting molecule on the particles. As it is now, it is difficult to assess whether delivery is due to specific or non-specific uptake into target cells, followed by subsequent cytoplasmic entry via endosomal escape mediated by the fusogen. Secondly, the importance of the antibody-based targeting molecule should be tested on cells that do not express GPC3 to assess uptake into non-target cells. This delivery strategy would still be valuable if eFT-CNVs favored delivery to target over bystander cells - the delivery doesn't need to be exclusive to GPC3-expressing cells, just preferential. However, if eFT-CNV delivery is equivalent between GPC3-positive and GPC3-negative cells, that would be a major limitation of this approach.

Points that should be addressed by the authors:

Membrane extrusion is an interesting strategy for scaling the production extracellular vesicle production. However, it seems possible that extruded membrane particles could be a mixture of particles, where some have the outer plasma membrane leaflet facing outward and some inward. Is there an experiment that the authors could do to get a sense of the proportion of particles that are displaying the targeting and fusogen molecules in the appropriate orientation?

More experimental detail should be provided on the membrane fragment extrusion process. The authors say that the supernatant was passed through 0.22- μm pore filter. Please provide more information on the filter. Was a mini extruder used? A vacuum filter?

The authors should add a citation to line 288-289 describing the fusogen.

The authors describe an experiment without showing the results (starting at 290). The data should be shown or the discussion of the result omitted.

Line 81/82 of the methods needs a citation.

Reviewer #1

To demonstrate the specificity, GPC3 rare expression or null expression cell lines should be used.

Thanks for the reviewer's suggestion. We supplied additional data to demonstrate the targeting/fusion specificity of various nanovesicles. In brief, GPC3^{KO} HepG2 cells and GPC3 expressing MCF7 cells were further tested. Cells were incubated with CNVs, eT-CNVs, and eFT-CNVs, respectively. The fluorescence images and quantitative data were shown in the updated Supplementary Fig. 3b and 3c. Nanovesicles were barely found in GPC3^{KO} HepG2 group. In MCF7 group, CNVs did not efficiently adhere to MCF7 cell membranes; eT-CNVs attached to MCF7 cell membranes through anti-GPC3 scFv but did not trigger membrane fusion; only eFT-CNVs fused to MCF7 cell plasma membranes and demonstrated dispersive green fluorescence. It is noteworthy that the average number of eT-CNVs on MCF-7 cell membranes was significantly lower in comparison with that on GPC3 overexpressing HepG2 cell membranes. Moreover, the average green fluorescence intensity of eFT-CNV treated MCF7 cells was lower than that in GPC3 overexpressing HepG2 cells. These findings were in line with the differential GPC3 expression level in two cell lines. The western blot of GPC3 derived from GPC3^{KO} HepG2, wild-type HepG2, and MCF7 cells was shown in updated Supplementary Fig. 3d. The supplied information was highlighted in the revised manuscript and Supplementary Information.

The author should discuss or show data to indicate whether the extrusion process change the cargos of the CNVs. What type of RNAs and proteins do they carry and are they therapeutic or toxic? Even HEK cells are widely used as an engineered cell line it will be great to check the cargos.

Thanks for the reviewer's concern. We agree with the reviewer's comments. It is critical to explore the differences

in cargo between EVs and CNVs. It is extremely important in EVs or CNVs based therapeutics which utilize the wrapped protein and RNA cargo to achieve treatment purposes. In our previous studies (*Cancer research, vol 70, 9371-9380, 2010*) we used western blot to compare membrane and cytosolic proteins of CNVs before and after drug loading. We posted the figures here for ease of reading (ENV: extracellular nanovesicles, they were naturally secreted EVs; AS1411-ENV: aptamer grafted CNVs). Our data showed that CNVs inherited these proteins during mechanical extrusion. But after drug loading with sonication, membrane-binding protein Annexin II and cytosolic proteins TSG101 and HSC70 were barely detectable in EVs and CNVs, indicating these proteins were depleted in drug loading. The shear stress generated by sonication can tear the lipid bilayer of EVs and CNVs. Subsequently, lipids can self-assemble and reform nanovesicles. In the repeated process of destruction and re-assembly, drugs were loaded, but the original protein cargo was almost depleted. We speculate that RNA cargo can also be efficiently depleted in drug loading. In another study (*ACS omega, vol 4, 22638-22645, 2019*) we used CNVs derived from human umbilical cord mesenchymal stem cells for skin rejuvenation. The results showed that CNVs can promote fibroblast migration, production of collagen, fibronectin and elastin, and wound healing. We did not observe the toxicity of CNVs. Peers also demonstrated that CNVs derived from stem cells can promote tissue regeneration. Relevant references can be found from this review paper (*Journal of nanobiotechnology, vol 19, 368, 2021*). These studies did not report the toxicity of CNVs, which may indicate that toxicity of CNVs is negligible. On the other hand, we loaded anti-cancer drugs into CNVs for cancer treatment. In comparison with intensely cytotoxic chemical compounds or toxins, the toxicity of residual proteins and RNAs derived from HEK293 cells can be ignored, if they are not fully depleted and if these residual proteins or RNAs are cytotoxic. Overall, the effect of residual proteins and RNAs after drug loading may have limited effects on recipient cells compared to anti-cancer drugs. Nevertheless, the exploration of residual proteins and RNAs was not the research aim of this study. It can be an independent study. Therefore, we will follow the reviewer's suggestion and investigate this concern in our future studies.

VSVG has also been proposed as a powerful tool to promote fusion. Please discuss the comparison between VSVG and Fusogen

Thanks for the reviewer's concern. In the manuscript we indeed discussed VSV-G. We highlighted this part in green in the manuscript. The relevant information was also pasted here as "*The existing fusogen expressing EVs, such as vesicular stomatitis virus G protein (VSV-G), a class III fusion protein, decorated EVs, only reach the tumor lesion through enhanced permeability and retention effect, a passive targeting strategy. Meanwhile, VSV-G EVs arbitrarily attach to both cancer and noncancerous cells through low density lipoprotein receptor, which is expressed in almost all tissues, and thus lead to poor specificity of drug delivery.*" The expression of low

density lipoprotein receptor in different normal tissues or tumors can be found through this link (https://www.cbioportal.org/results/plots?cancer_study_list=ccl_broad_2019&Z_SCORE_THRESHOLD=2.0&RPPA_SCORE_THRESHOLD=2.0&profileFilter=mutations%2Cstructural_variants%2Ccna&case_set_id=ccl_broad_2019_cnaseq&gene_list=LDLR&geneset_list=%20&tab_index=tab_visualize&Action=Submit&plots_horz_selection=%7B%22selectedGeneOption%22%3A2719%2C%22dataType%22%3A%22clinical_attribute%22%2C%22selectedDataSourceOption%22%3A%22CANCER_TYPE_DETAILED%22%7D&plots_vert_selection=%7B%22selectedGeneOption%22%3A2719%2C%22dataType%22%3A%22MRNA_EXPRESSION%22%2C%22selectedDataSourceOption%22%3A%22rna_seq_mrna%22%2C%22logScale%22%3A%22true%22%7D&plots_coloring_selection=%7B%7D). In comparison, GPC3 expression in normal tissues is very low (https://www.cbioportal.org/results/plots?cancer_study_list=ccl_broad_2019&Z_SCORE_THRESHOLD=2.0&RPPA_SCORE_THRESHOLD=2.0&profileFilter=mutations%2Cstructural_variants%2Ccna&case_set_id=ccl_broad_2019_cnaseq&gene_list=GPC3&geneset_list=%20&tab_index=tab_visualize&Action=Submit&plots_horz_selection=%7B%22selectedGeneOption%22%3A2719%2C%22dataType%22%3A%22clinical_attribute%22%2C%22selectedDataSourceOption%22%3A%22CANCER_TYPE_DETAILED%22%7D&plots_vert_selection=%7B%22selectedGeneOption%22%3A2719%2C%22dataType%22%3A%22MRNA_EXPRESSION%22%2C%22selectedDataSourceOption%22%3A%22rna_seq_mrna%22%2C%22logScale%22%3A%22true%22%7D&plots_coloring_selection=%7B%7D).

B2M was knocked out. However, it is unclear whether B2M-deficient CNV can significantly promote biocompatibility. It is a clever strategy to use CRISPR technology to remove MHC and GPC3. Can the authors show actually the resulted cells and CNVs are less immunogenic? The field has some debating on the immunogenicity of EVs and CNVs. The current notion is that while allogeneic cells are immunogenic, allogeneic EVs or CNVs are not that immunogenic. It will be crucial if the authors can demonstrate that this removal of MHC and GPC3 are necessary and essential.

Thanks for the reviewer's concern. The knockout of GPC3 is to avoid potential agglomeration of cells and CNVs. HEK293 cells express a small amount of GPC3 on membranes to maintain normal functions in cellular signaling including cell growth (*Nature Communications*, vol 6, 6536, 2015). If we directly overexpress anti-GPC3 scFv on GPC3 expressing HEK293 cells, the intercellular interaction between GPC3 and scFv may induce cell agglomeration. The agglomeration would be worse among CNVs which harbor both GPC3 and scFv on the membranes. For example, in Fig. 3g we demonstrated that HepG2-derived EVs can interact with scFv expressing CNVs and form nanovesicle clusters. Therefore, we knocked out GPC3 of HEK293 cells to exclude self-agglomeration of CNVs.

In regard to the knockout of MHC-I and necessity, please allow us to introduce the rationale step-by-step. (1) MHC-I knockout enables immune evasion. It is a known fact that allogeneic cells can induce significant immune responses in recipients. Therefore, in CAR-T therapy and stem cell therapy, for examples, autogenic cells are preferred. However, in comparison with the massive number of allogeneic cells donated by healthy volunteers, autogenic CD8 T cells or stem cells may not always available. To simultaneously address the shortage concerns and unfavorable immune responses, universal CAR-T cells and universal stem cells have been developed, which

was achieved by permanent knockout of MHC-I (*Cell Research*, vol 27, 154–157, 2017; *Journal of the American Heart Association*, vol 7, e010239, 2018). The knockout of MHC-I actually was fulfilled by the knockout of B2M, a subunit of MHC-I. Without B2M the incorrect MHC-I can be recycled and destructed. Without MHC-I cells cannot present antigens on cell membranes to CD8 T cells, achieving immune evasion. (2) Cell derived EVs carry MHC molecules, and these MHC molecules still can present antigens to recipient cells. For example, dendritic cell derived EVs can present antigen through membrane MHC-II (*Annual Review of Immunology*, vol 36, 435-459, 2018). Nucleated cell derived EVs can also present antigen through membrane MHC-I (*Journal of Immunology*, vol 183, 1884–1891, 2009). (3) In cell culture, aging cells and apoptotic cells are inevitable. MHC-I presents primarily endogenous antigens. The abnormal proteins derived from aging cells or cellular debris from apoptotic cells can be presented by MHC-I on EVs and further induce immune response. Based on the abovementioned evidence, we speculate that MHC-I knockout can further improve biocompatibility of EVs. (4) Notably, it might be very challenging to demonstrate that MHC-I deficient EVs are more biocompatibility than EVs with MHC-I. The difference in immunogenicity between the two would be small if aging or apoptosis does not exist on a large scale. (5) Lastly, when we knocked out the intrinsic GPC3 we incidentally got MHC-I knocked out. Double knockout with CRISPR/Cas9 does not increase the cost or experimental complexity.

Regarding the debate of high immunogenicity of CNVs over EVs, it might be reasonable. Cell debris generated during the mechanical extrusion and the exposed cytosolic proteins could be presented through MHC-I on CNV membranes. In comparison, naturally secreted EVs have less chance to present debris or cytosolic proteins given most of donor cells are healthy. This speculation may illustrate the need for MHC-I knockout. The other possibility is that the flip of inner plasma membrane may induce immune response. When the positively charged lipids and integral proteins on inner membranes flipped during mechanical extrusion and self-assembly of CNVs, these CNVs will trigger phagocytic clearance and immune response. But CNVs with positively charged surface can aggregate with CNVs with negatively charged surface, and thus the majority of them can be removed during 16,500 g centrifugation and subsequent filtration. In the updated Supplementary Fig. 4b, we demonstrated the average zeta-potential of our nanovesicles was negative. Moreover, in our previous studies we did not observe significant immunogenicity of CNVs (*Cancer research*, vol 70, 9371-9380, 2010; *ACS omega*, vol 4, 22638-22645, 2019; *Bioactive materials*, vol 6, 749-756, 2021; *Bioactive Materials*, vol 9, 251-265, 2022).

The relevant discussion was added into the revised manuscript and highlighted in yellow.

Reviewer #2

(Expression level and linker length) About scFv and fusogen expression, expression level and linker length from plasma membrane might possibly affect the molecular targeting and membrane fusion efficacy. Optimization should be needed.

Thanks for the reviewer's concern. We agree with the reviewer's comment that the linker length may influence function of membrane protein due to potential steric effects. It is noteworthy that the expression of membrane-bound antibody and fusogen on cell membranes have been investigated for decades (*Nature Medicine*, vol 11, 346-352, 2005; *PNAS*, vol 103, 11479-11484, 2006; *Journal of tissue engineering and regenerative medicine*, vol 4, 247-258, 2010; *Nanoscale*, vol 10, 14230-14244, 2018; *International journal of nanomedicine*, vol 14, 8755-8768, 2019; *Theranostics*, vol 9, 5657-5671, 2019). Recently, in the field of EV peers constructed cell strains which overexpress membrane-bound targeting moieties and then harvested disease targeting EVs for therapeutics. More information can be found from this review paper (*Signal transduction and targeted therapy*, vol 8, 124, 2023). In brief, expression membrane-bound protein is a very mature technique, and commercialized pDisplay plasmids are available everywhere. Numerous studies demonstrated that without linkers membrane-bound targeting moieties, including antibody, scFv, peptide, and nanobody, work properly. Previous studies also demonstrated that membrane-bound VSV-G without a space linker work properly (*Nature Medicine*, vol 11, 346-352, 2005; *PNAS*, vol 103, 11479-11484, 2006; *International journal of nanomedicine*, vol 12, 3153-3170, 2017); *Advance materials*, vol 29, 1605604, 2017; *Advanced functional materials*, vol 33, 2209056, 2023). Therefore, we would like to follow the well-established protocol and use well-demonstrated commercial kits to construct our cell strain. Going back to the reviewer's comments, to systemically optimize the space linker we need to consider not only the length, but also the rigidity and surface charge. Subsequently, other reviewers and readers may request more studies to clarify that whether these linkers influence the bioactivity, expression, folding, stability, and pharmacokinetics of these membrane fusion proteins (*Advanced drug delivery reviews*, vol 65, 1357-1369, 2012). These studies might be extremely time consuming and even make the study endless. Overall, at the current stage supplement and optimization of a space linker to the fusion protein are not compulsory or imperative.

Regarding the expression level of scFv and fusogen, we used PCR to detect the relative mRNA expression level and found that "the relative mRNA expressions of anti-GPC3 scFv and fusogen were 237,736 and 1,867 copies." Moreover, we used anti-HA Fab-modified immune gold nanoparticles to detect the number of fusogen on eFT-CNV membranes and found that average 7 fusogens on an eFT-CNV. In the revised manuscript this information was highlighted in green for ease of reading. We are unable to directly use immunostaining to detect the number of anti-GPC3 scFv on cell membranes or eFT-CNV membranes because relevant antibody targeting anti-GPC3 scFv is unavailable. That is why that the number of membrane-bound antibodies were not quantitatively detected in previous studies (*Frontiers in immunology*, vol 7, 690, 2017; *Nature Medicine*, vol 11, 346-352, 2005; *PNAS*, vol 103, 11479-11484, 2006; *Journal of tissue engineering and regenerative medicine*, vol 4, 247-258, 2010; *Nanoscale*, vol 10, 14230-14244, 2018; *International journal of nanomedicine*, vol 14, 8755-8768, 2019; *Theranostics*, vol 9, 5657-5671, 2019).

(Negative control about non-expression of GPC3) There was no data *in vitro* (cellular uptake and biological activity) and *in vivo* (solid tumor targeting) about the case of non-expression of GPC3 on cells as negative control. In addition, how about only fusogen expression (without scFv expression) on CNVs in each experiment??

Thanks for the suggestion. The reviewer 1 raised the same concern. Please refer to the response there. Briefly, we supplied additional data to demonstrate the targeting/fusion specificity of various nanovesicles. GPC3^{KO} HepG2 cells and GPC3 expressing MCF7 cells were further tested *in vitro*. The new data in combination with Fig. 3f demonstrated that membrane fusion also relies on GPC3 expression level because our engineered fusogen is binding-defective but fusion-competent. The eFT-CNVs need to utilize anti-GPC3 scFv to anchor onto cell membranes and then get the subsequent membrane fusion done efficiently via fusogens.

Without the assistance of anti-GPC3 scFv, *i.e.*, eF-CNVs (the reviewer mentioned in the comment; equipped with fusogens but without anti-GPC3 scFv) cannot efficiently and firmly anchor onto cell membranes. Without firm interaction between eF-CNVs and target cells, fusogens do not have sufficient time to disassemble E1-E2 heterodimers, project E1 outwards, leisurely insert into target cell membranes, and trigger membrane fusion. For example, in tumor microenvironment eF-CNVs have a small chance to attach onto tumor cells due to poor interaction between eF-CNVs (without scFv) and tumor cells. Meanwhile, body fluid produced shear forces can easily take eF-CNVs away from tumor cell surface. Therefore, membrane fusion is unlikely to happen. In exceptional circumstances, membrane fusion can happen without targeting moieties. For example, eF-CNVs are physically trapped on tumor cell membranes, and membrane fusion can finally happen in the acidic tumor microenvironment. In brief, the anti-GPC3 scFv provides cancer targeting effects, but more importantly, scFv can anchor eFT-CNVs on cell membranes, allowing membrane fusion can be successfully completed. Therefore, binding-defective but fusion-competent fusogen must work with anti-GPC3 scFv to specifically and efficiently finish cytosolic drug delivery.

The reviewer might further be wondering that why the authors did not construct a HEK293 cell strain with membrane bound fusogens only (eF-HEK293) and then harvest derived CNVs, *i.e.*, eF-CNVs, for drug delivery. Such construction does not actually benefit our investigation as we analyzed in the last paragraph. In addition, the differential expression of fusogen on membranes may lead to inconsistent comparisons. In this study, we expressed anti-GPC3 scFv first and then expressed fusogens on HEK293 membranes (version 1). Based on PCR analysis, we found 237,736 and 1,867 copies of scFv and fusogen mRNAs. We can express fusogens first and harvest eF-HEK293 cells and derived eF-CNVs. Subsequently, we further let eF-HEK293 cells express anti-GPC3 scFv and construct the other eFT-HEK293 cell strain (version 2). The major concern is that the number of expressed fusogens and scFv on cell membranes will be certainly different between the version 1 and the version 2 HEK293 cell strains. The difference would further induce inconsistent comparison among CNVs, eT-CNVs (harvested from version 1 engineered HEK293 cells), eF-CNVs (harvested from version 2 engineered HEK293 cells), and eFT-CNVs (harvested from both version 1 and 2 engineered HEK293 cells). Such experimental design would be complex and chaotic. Given a few fusogens are sufficiently for membrane fusion once eFT-CNVs firmly anchor onto tumor cell membranes. Therefore, we overexpressed anti-GPC3 scFv (to enhance interaction odds and binding stability) and then express fusogens. We did not construct eF-HEK293 cells additionally.

Because the negative findings were identified in the group of GPC3^{KO} HepG2 cells we did not further perform the relevant animal studies with GPC3^{KO} HepG2 tumor. First, peers very rarely use targeting moieties functionalized nanoparticles (e.g., eFT-CNVs in this study) to treat tumors without the cognate molecules on tumor cell membranes (e.g., GPC3^{KO} HepG2 tumor in this study). Such experimental design may easily lead to negative findings. The readers would further wonder if the result was a true negative or a false negative. It is difficult to answer this tricky question. Second, mouse cells do not express human GPC3 on cell membranes, and thus mouse cells naturally serve as negative controls *in vivo*. In updated Supplementary Fig. 8c, we did not observe significant toxicity in mouse heart, spleen, lung, and kidney. Limited cytotoxicity was observed in mouse liver, simply because liver can heavily and nonspecifically uptake these anti-cancer drug loaded eFT-CNVs. The animal data indeed demonstrated the specificity of eFT-CNVs *in vivo*. Third, if we use eFT-CNVs to treat GPC3^{KO} HepG2 tumor *in vivo*, we only investigate the treatment effect of passive targeting. But passive targeting effect has been investigated in the group of HepG2 cells treated with drug loaded CNVs (Fig. 3h). eFT-CNVs cannot efficiently bind to GPC3^{KO} HepG2 cell membranes, and without membrane-bound scFv CNVs cannot efficiently bind to HepG2 cell membrane either. Essentially, there is not difference between the two scenarios. Forth, in our animal study we already used PBS, CNV, and eT-CNV as three negative controls. These controls were adequate and demonstrated the specificity of eFT-CNVs *in vivo*.

(Supplementary Figure 2) What are “tfHEK293” and “HepG2 GPC3 Blk”?? In addition, detailed data of “only binding on cell surface” and “membrane fusion” should be assessed in each condition.

Thanks for the reviewer’s concern and suggestion. In the updated Supplementary Fig. 2, we revised “tfHEK293” to “eFT-HEK293”. We used “tfHEK293” in the previous version of our manuscript and forgot to update them in Supplementary Fig. 2. “HepG2 GPC3 Blk” was revised as “HepG2 GPC3^{blocked}”. In the manuscript, we mentioned anti-GPC3 antibodies were used to block GPC3 on HepG2 cell membranes. In the updated Supplementary Fig. 3c, we counted the fluorescent dots (PKH67 labeled nanovesicles) in the group of CNV and eT-CNV, respectively. In the group of CNV, fluorescent dots were barely detected. In the group of eT-CNV treated GPC^{KO} HepG2, HepG2, and MCF7 cells, we detected average 8.3 dots on HepG2 cell and 3.7 dots on MCF7 cells (ANOVA, $p < 0.05$). In the group of eFT-CNV, the average green fluorescence intensity of 200 HepG2 cell and MCF7 cells was 22.4 a.u. and 7.8 a.u. (ANOVA, $p < 0.05$). Without cognate GPC3 on cell membranes, we barely detect fluorescent dots or dispersive green fluorescence on GPC^{KO} HepG2 cell membranes.

(Figure 2e and 3f) The experiment for checking cell-cell fusion in only pH5.5 condition. How about pH7 condition?? Negative control experiments should be needed. In addition, how about HEK293 without any expression of scFV and fusogen??

Thanks for the reviewer’s concern. Sindbis virus fusion protein requires acidic environment to finish membrane fusion (*Journal of virology*, vol 74, 5667-5678, 2000). In the manuscript, we already discussed that “The E2 has a cell recognition function, whereas E1 facilitates fusion of the viral and cellular membranes, which requires an acidic environment and a cholesterol membrane component.” and “We are thankful for previous studies on the optimization of the pH for membrane fusion with excellent repeatability”. The reviewer’s concern has been well

investigated by previous studies. We do not need to repeat it in this study. Actually, in the discussion we further mentioned that “Lastly, not all diseases provide an acidic condition like a tumor microenvironment, which allows pH-dependent fusogen to work efficiently. Therefore, to widely apply this drug delivery system, the pH-independent fusogens are highly desired.”.

The eT-HEK293 cells without fusion competent served as a negative control. Moreover, in the Supplementary Fig. 2 we demonstrated that HEK293 without scFv and fusogen as another negative control did not interact with HepG2 cells. Regarding the reviewer’s concern on Fig. 3f, we supplied data collected from GPC3^{KO} HepG2 and MCF7 cells in Supplementary Fig. 3b and 3c.

(Figure 4b) Delivery of siR-Sox2 using eFT-CNVs, how about off-target effects??

Thanks for the reviewer’s concern. Detailed explanation can be found in the response to the second comment. The data collected from GPC3^{KO} HepG2 cells, HepG2 cells, and MCF7 cells demonstrated that the drug delivery efficiency and specificity of eFT-CNVs. The data reflected that the off-target effect of eFT-CNVs was very limited *in vitro*. Once again, we want to kindly remind the reviewer that the engineered fusogen is binding-defective but fusion-competent. The eFT-CNVs rely on anti-GPC3 scFv to efficiently attach onto GPC3 expression cell membranes. The treatment efficacy and specificity depend on GPC3 expression. Moreover, to successfully complete cytosolic delivery, an acidic microenvironment is also required. If cells do not express GPC3 on membrane or without an acidic microenvironment, eFT-CNVs barely attach to these cells or cannot complete membrane fusion. Notably, normal cell only express a small amount of GPC3 to maintain normal functions in cellular signaling including cell growth (*Nature Communications*, vol 6, 6536, 2015). The expression level of GPC3 in normal tissue can be found via the link below. In addition, healthy tissues normally do not form an acidic microenvironment either. Therefore, the side-effect of eFT-CNVs to normal tissues is also expected to be very limited *in vivo*. The animal data supported our claim. We did not observe significant toxicity in mouse major organs because mouse cells do not express human GPC3 on membranes.

(https://www.cbioportal.org/results/plots?cancer_study_list=ccl_broad_2019&Z_SCORE_THRESHOLD=2.0&RPPA_SCORE_THRESHOLD=2.0&profileFilter=mutations%2Cfusion%2Ccna&case_set_id=ccl_broad_2019_cnaseq&gene_list=GPC3&geneset_list=%20&tab_index=tab_visualize&Action=Submit&plots_horz_selection=%7B%22selectedGeneOption%22%3A2719%2C%22dataType%22%3A%22clinical_attribute%22%2C%22selectedDataSourceOption%22%3A%22CANCER_TYPE_DETAILED%22%7D&plots_vert_selection=%7B%22selectedGeneOption%22%3A2719%2C%22dataType%22%3A%22MRNA_EXPRESSION%22%2C%22selectedDataSourceOption%22%3A%22rna_seq_mrna%22%2C%22logScale%22%3A%22true%22%7D&plots_coloring_selection=%7B%7D).

On the other hand, the specificity of anti-GPC3 scFv has been well demonstrated by previous studies. The development of this anti-GPC3 scFv can be traced back almost 20 years. Peers used this anti-GPC3 scFv for immunotherapy or use it to construct CAR-T cells for immunotherapy. The inventors also have several patents of this anti-GPC3 scFv (*Frontiers in immunology*, vol 7, 690, 2017; *Cancer research*, vol 64, 2418-2423, 2004; *Biochemical and Biophysical Research Communications*, vol 378, 279-284, 2009; Patent application number: PCT/US2015/031997 and PCT/JP2003/011320).

(Figure 4d and 4e) In Figure 3f, eFT-CNVs showed much higher fusion than that of eT-CNVs. However, in Figure 4d and 4e, effective degree of fugogen expression for biological activity was not so high (ex. PTX: eFT-CNV 10.9, eT-CNV 18.2). Reasons why? Fusogen did not worked well in the expression design, or not??

Thanks for the reviewer's concern. In Fig. 3f, nanovesicles were incubated with HepG2 cells for only 30 minutes. The timescale was optimized in our previous studies (*Cancer research*, vol 70, 9371-9380, 2010; *Bioactive materials*, vol 6, 749-756, 2021; *Bioactive Materials*, vol 9, 251-265, 2022). If the incubation time is too short, e.g., 5 minutes, even eT-CNVs or eFT-CNVs could not quickly diffuse to and anchor onto cell membranes. If the incubation time is too long, e.g., 48 hours, dye labeled CNVs and eT-CNVs may escape from lysosomes. The dissociated lipid dyes from CNVs or eT-CNVs can integrate into cellular plasma inner membranes and show fluorescent signals. Therefore, we only incubated nanovesicles with HepG2 cells for 30 minutes to clearly show the effect targeting and membrane fusion functions of eFT-CNVs. It explains why eFT-CNVs showed much higher fusion effect. In sum, Fig. 3f only demonstrated the membrane binding and fusion effect of nanovesicles in 30 minutes. In Fig 4d and 4e, cells were incubated with nanovesicles for 48 hours. The majority of CNVs and eT-CNVs were taken up through endocytosis in 48 hours. Notably, drugs loaded in CNVs and eT-CNVs will not be completely destroyed in lysosomes. Approximately 30%-40% drugs can escape from lysosomes and enter into cytosol (*Journal of cell biology*, vol 213, 173-184, 2016; *Nature reviews molecular cell biology*, vol 19, 213-228, 2018). The lysosomal escape allows CNVs and eT-CNVs to deliver drugs to cytosol and fulfill the treatment tasks. Although the drug delivery efficacy of CNVs and eT-CNVs is lower than that of eFT-CNVs, they still have treatment effect to a certain degree. In brief, there is no linear relationship between binding/fusion efficiency and treatment efficacy in cells. Without membrane fusion, limited CNVs and eT-CNVs still can deliver drugs to cytosol through lysosomal escape and thus demonstrate treatment efficacy. The duration of the experiments in Fig. 3f and Fig. 4d-4e was significantly different. Therefore, it is not appropriate to compare the binding/fusion efficiency at 30-min timepoint to the treatment efficacy at 48-h timepoint.

(Figure 4f) Each result of "free drug" in CNC, eT-CNV, and eFT-CNV was very different. Reasns why?

Thanks for the reviewer's concern. The midnight blue color indicates free drugs. It could be siRNA, gelonin, or paclitaxel. The column 2-5 were free siRNA (midnight blue), siRNA loaded CNV (green), siRNA loaded eT-CNV (yellow), and siRNA loaded eFT-CNV (red). The column 6-9 and column 10-13 were gelonin and paclitaxel related. The treatment efficacy was different among free siRNA (column 2), free gelonin (column 6) and free paclitaxel (column 10) due to different therapeutic potency.

(Supplementary Figure 4) In the experiment of molecular release from each CNV in pH5.5 or pH7.4, pH5.5

condition showed higher release efficacy than that of pH7.4 condition. Reasons why about pH-dependency? In the case of liposome, pH-dependent release will be also shown, or not?? In addition, retention efficacy of bioactive molecules in CNVs (derived from plasma membranes) and liposome is similar, or not??

Thanks for the reviewer's concern. The original Supplementary Fig. 4 was updated as Supplementary Fig. 5. Similar finding was reported by peers (*International journal of nanomedicine*, vol 14, 8603-8610; *Journal of biological chemistry*, vol 284, 34211-34222, 2009; *RSC Advances*, vol 10, 28314-28323, 2020). The exact mechanism is unknown. In the manuscript we briefly mentioned that "This may be attributed to the relative instability of nanovesicles at low pH.". To understand this phenomenon, additional studies are required to investigate the molecular mechanisms. It is reasonable to speculate that the abundant hydrogen ions in solution may impair the weak van der Waals attractive forces between lipids, making CNVs unstable. Nevertheless, the relevant mechanisms are not our main research topic and may make the relevant studies endless. In this study, we need to focus on targeting moiety and fusogen co-mediated cytosolic drug delivery.

In the liposomal formulation, low pH may promote drug release from regular liposomes (*PharmSciTech*, vol 15, 845-857, 2014). Peers also can make liposomes very stable at acidic environments (*Advanced intelligent system*, 1900124, 2020). On the contrary, acidic pH-responsive liposomes are more popular in cancer treatment. Over the decades peers have been investigating pH-responsive liposomes which can respond to the acidic tumor microenvironment and rapidly release drugs in the tumor microenvironment. More information can be found in this review paper (*Frontiers in Oncology*, vol 12, 855019, 2022).

We did not understand the third question the reviewer raised here. We did not see that why we need to investigate the retention efficacy of payload between CNVs and liposomes. They are two different materials. First, there are hundreds and thousands of liposomal formulations. There is not standard liposome for such comparison. Second, CNVs and liposomes have different properties. CNVs are cell derived nanovesicles which are decorated with membrane-bound proteins. Liposomes are artificially prepared without membrane-bound proteins, and normally PEGylated liposomes are frequently used for drug delivery. Third, liposome is not our research aim in this study. Therefore, we are unable to answer this question.

(Cellular uptake mechanism) Cellular uptake mechanism of the eFT-CNVs via GPC3 (glypican 3)??

Thanks for the reviewer's concern. Once again, we want to kindly remind the reviewer that eFT-CNVs can complete cytosolic drug delivery via fusogen mediated membrane fusion. It was not based on GPC3 mediated endocytosis. In the scenario of eFT-CNVs, GPC3 on the cell membrane attracts eFT-CNVs and further anchors eFT-CNVs through the membrane bound anti-GPC3 scFv. The firm attachment of eFT-CNVs in combination with the acidic environment can trigger membrane fusion and complete cytosolic drug delivery. Therefore, GPC3 based cellular uptake mechanism was not our research aim.

On the other hand, we do not really know how far the reviewer wanted us to dig into the mechanisms. It could be very superficial if we just use routine endocytosis inhibitors to quickly check which type of internalization can be inhibited. However, repeated investigation of these inhibitors and relevant molecules does not expand our knowledge. For example, it is well known that receptor-mediated endocytosis is called clathrin-mediated endocytosis (*Nature reviews molecular cell biology*, vol 19, 313-326, 2018). Clathrin involves in the antibody-

GPC3 complex mediated endocytosis. Alternatively, the deep exploration of the mechanisms of GPC3-mediated endocytosis will expand our knowledge, but the relevant studies will be endless and deviate from our research topic. Over the decades, many molecules involved in GPC3-mediated endocytosis were identified. More details can be found in these selected references (*Matrix biology, vol 35, 51-55, 2014; Biochemical journal, vol 410, 503-511, 2008; Oncology letter, vol 20, 1-11, 2020; Journal of cell science, vol 125, 3380-3389, 2012; Frontiers in oncology, vol 9, 708, 2019*). So far, the mechanisms were not fully understood yet. It is not realistic to expect us clearly this question in an engineering & technology research.

Nevertheless, to meet the reviewer's curiosity we used PKH67 labeled eT-CNVs rather than eFT-CNVs to investigate the potential mechanisms. The data was supplied in Supplementary Fig. 3e and 3f. Chlorpromazine, genistein, and cytochalasin D did inhibit the uptake of eT-CNVs. Therefore, clathrin-mediated endocytosis, micropinocytosis, and caveolae-mediated endocytosis were involved in the internalization process of eT-CNVs. The relevant experimental description and findings were updated in the revised manuscript and Supplementary Information. Please keep in mind that this is a superficial mechanism of GPC3-mediated endocytosis and cannot fully represent the process in the case of GPC3-scFv interaction triggered endocytosis.

(Supplementary Figure 7) How about concentration dependency of each CNVs about biological anti-tumor activity in vivo?? In addition, in Figure 7C, negative control in each organ should be added. Data of body weight should be also added for assessing side effects as a supplementary figure.

Thanks for the reviewer's suggestion. High drug dose may improve anti-cancer treatment efficacy. Meanwhile, the high dose also puts mice at risk for drug-related toxicities. If mice die during the treatment, it may lead to the experimental failure. IACUC may also require detailed explanation. Moreover, high dose may lead to insignificant difference in treatment efficacy between cancer-targeted drug delivery and free drugs. Wrong conclusion may be drawn. On the contrary, low dose may not have any treatment efficacy, which may also lead to wrong conclusions. It is highly controversial to have high, middle, and low dose groups in experiments. Peers normally do not investigate the treatment efficacy of various dosage. For example, in this Nature paper (*Nature, vol 546, 498-503, 2017*) the investigators selected optimal dose first and then launched the animal studies. In fact, peers need to follow the guidelines and determine the reasonable dose before animal studies (*Journal of basic and clinical pharmacy, vol 7, 27-31, 2016*). Normally, based on equivalent dose, shape factor, volume of dosage, and other factors, peers can determine the optimal drug dose before the experiments. Based on the drug type, tumor type, and animal type, peers also can refer the published references and find out the frequently used doses. In the section of drug loading and characterization, we already determined the average amount of drugs loaded into nanovesicles. Based on the optimal drug dose and drug loading efficiency, we prepared drug loaded nanovesicles, ensuring that 1 mg siRNA-Sox2-, 2.5 mg gelonin-, and 7.5 mg of paclitaxel-equivalent per kg of

body weight per dose will be injected in each subgroup.

In the previous version of our manuscript, due to limited space we did not include the morphology of tissues derived from the PBS negative control group. In the updated Supplementary Fig. 8, we reduced the size of Supplementary Fig. 8a and supplied morphology of tissues collected from PBS group in Supplementary Fig. 8c. The body weight data was added as Supplementary Fig. 9.

(Supplementary Figure 8) eFT-CMV enhanced retention not only in tumor but also in each organ. Detailed assessment about damage in each organ should be needed.

Thanks for the reviewer's concern. We regret that we are unable to fully recover the data in this section. This whole study was finished in last August, almost seven months ago. The tissue samples have been discarded following the protocol of animal study. As the reviewer noticed that we had three groups (siRNA, gelonin, and paclitaxel). Each group has five subgroups (PBS, free drug, drug@CNVs, drug@eT-CNVs, and drug@eFT-CNVs). In each subgroup, heart, liver, spleen, lung, kidney, and tumor tissues need to be collected. Each tissue requires at least five slides for histological analysis. It means that we need to prepare at least $3 \times 5 \times 6 \times 5 = 450$ tissue slides, take at least 2,250 pictures, and finally present representative 90 pictures here. The workload is extremely overwhelming with very limited value. Therefore, we only investigated the morphology of major organs (heart, liver, spleen, lung, and kidney) in eFT-CNV subgroups, given eFT-CNV was the main subject of this study. Based on the same reasoning, we only investigated pharmacokinetics (PK) of paclitaxel in eFT-CNV subgroup at 24-h timepoint. Through the comparison of tissue morphology among four groups (NC, siRNA@eFT-CNVs, gelonin@ eFT-CNVs, and paclitaxel@eFT-CNVs), we did not observe significant changes in major organs with the exception of livers. siR-Sox2, gelonin, and paclitaxel loaded eFT-CNVs demonstrated hepatotoxicity. The main reason is the Kupffer cells and macrophages in liver can non-specifically and heavily uptake nanoparticles. Without CD47 decoration on the outer membrane, liposomes, EVs, and CNVs always encounter this first-pass

effect (*ACS Nano*, vol 13, 3522–3533, 2019; *Frontiers in pharmacology*, vol 6, 2862015; *Nature*, vol 546, 498-503, 2017; *Science*, vol 339, 971-975, 2013). Notably, all mice in the experiments survived. The body weight of mice (Supplementary Fig. 9) reflected that the hepatotoxicity or systemic toxicity was limited.

Moreover, in this study we already had 75 mice for investigation of treatment efficacy (siRNA, gelonin, and paclitaxel) and another 10 mice for investigation of PK at the 24-h timepoint. Because we do not have dedicated funds for this study, we have to give up some less important investigations. To sum up, due to overwhelming workloads and limited budget, we mainly investigated the treatment efficacy and systemic toxicity of drug loaded eFT-CNVs. Thanks for your understanding.

(Western blot) There were only objective molecular size data in the manuscript. Full molecular size range data should be added as supplementary figures.

Thanks for the reviewer's concern. In western blot, we used protein marker to indicate the position of target proteins. Because of the experimental habits, the first author did not purposely acquire both white light and chemiluminescent pictures. Any, please find the attached pictures below. The grey arrows indicate the trace of protein markers. We do not usually use an intact PVDF membrane with full-range markers to show the target protein size. Instead, based on the protein marker indicated position we cut a small piece of PVDF membrane which contains our target protein for subsequent antibody labeling. This operation is also very common. The main reason is that the intact PVDF membrane requires 5-10 times more monoclonal antibody for protein detection compared to the antibody usage in detection with a small piece of PVDF. It is such a waste of antibodies. The membrane cutting saves our antibody usage, but the cutting cannot provide full-range markers in Fig. 3c and 3e or cause the information loss in Fig. 2c. Nevertheless, we supplied the molecular weight of our target proteins in updated Fig. 2c, Fig. 3c, and Fig. 3e. The original western blot pictures were further provided in the updated Supplementary Information.

GAPDH: 37 kDa (marker: Thermofisher 26616)

marker: 55, 40, and 35 (from top to down)

CD81: 22 kDa (marker: Thermofisher 26616)

marker: 25 and 15 (from top to down)

TSG101: 50 kDa (marker: Thermofisher 26616)

marker: 55

E2 domain harboring a HA tag: ~49 kDa (marker: Thermofisher 26616)

marker 55 and 40 (from top to down)

To further confirm the size of E2 domain harboring a HA tag we used another protein marker (BioRad,

1610375). Lanes from left to right are eFT-CNVs, eT-CNVs, CNVs, HepG2, eFT-CNV treated HepG2, respectively. It shows that the size of E2 with a HA tag is close slightly less than 50 kDa.

marker: 100, 75, 50, 37, and 25 (from top to down)

(Figure 2d) What are X-axis and Y-axis??

Thanks for the reviewer's concern. X-axis: FL1-H AF488; Y-axis: ssc-H. We updated Fig. 2d in the revised manuscript.

(Supplementary Figure 3) siR-Sox2, gelonin, and paclitaxel were showed to affect zeta-potential. There are high amount of surface binding on vesicular membrane (not encapsulation), or not??

Thanks for the reviewer's concern. siR-Sox2 may physically attach onto nanovesicle membrane proteins. These siRNAs on nanovesicle membranes cannot fully removed unless we use RNases. In drug delivery, the membrane bound siRNAs will be digested by nucleases in body fluid or destroyed in lysosomes after endocytosis. In the scenario of membrane fusion, these siRNAs may stay on target cell membranes.

Negatively charged small-molecule paclitaxel may have small chances to attach onto negatively charged nanovesicles membranes. But we cannot exclude that paclitaxel may interact with membrane proteins, and thus stay on the outer membranes.

Regarding positively charged gelonin, we do not think gelonin-bound nanovesicles can stably exist in the suspension. The positively gelonin will trigger aggregation of nanovesicles via electrostatic interaction. The aggregation of nanovesicle can directly ruin the subsequent cell or animal treatment. However, we did not encounter this problem. Moreover, zeta potential of gelonin-bound nanovesicles would be positive. The surface attachment of gelonin would influence the value of zeta potential. But the detected zeta potential was negative in all three groups.

Overall, we admit that siRNA and paclitaxel may attach onto nanovesicles outer membranes. Compared to the amount of drugs loaded into nanovesicles, the amount of membrane-bound drugs would be very limited and would not significantly influence the downstream investigations. Therefore, peers never ever intentionally removed the membrane bound drugs in numerous studies. Normally, after drug loading either ultracentrifugation or filtration will be used to remove unloaded drugs. The harvested drug-loaded nanovesicles will be directly used for treatment. On the other hand, removing the membrane bound drugs is challenging. Although the amount of membrane-bound drugs can be decreased after repeated purification, the yield of drug-loaded nanovesicles will be significantly impaired, making the downstream investigation impossible.

(Supplementary Figure 5 and 6) What are "negative controls" of gelonin and paclitaxel??

Thanks for the reviewer's concern. Supplementary Fig. 5 & 6 were updated as Supplementary Fig. 6 & 7. In groups of siR, gelonin, and paclitaxel, we consistently used PBS, drug loaded CNV, and drug loaded eT-CNVs as negative controls. Free drug was used as a positive control.

Reviewer #3

It is difficult to assess the cell-type-specific delivery of eFT-CNVs in the experiments presented by the authors. For example, it would have greatly improved the manuscript if the authors had included an extruded CNV displaying a fusion molecule in the absence of a targeting molecule. That critical control particle would allow for the interpretation of the importance of displaying an antibody-derived targeting molecule on the particles. As it is now, it is difficult to assess whether delivery is due to specific or non-specific uptake into target cells, followed by subsequent cytoplasmic entry via endosomal escape mediated by the fusogen. Secondly, the importance of the antibody-based targeting molecule should be tested on cells that do not express GPC3 to assess uptake into non-target cells. This delivery strategy would still be valuable if eFT-CNVs favored delivery to target over bystander cells - the delivery doesn't need to be exclusive to GPC3-expressing cells, just preferential. However, if eFT-CNV delivery is equivalent between GPC3-positive and GPC3-negative cells, that would be a major limitation of this approach.

Thanks for the reviewer's suggestion. The reviewer 1 raised the same concern. In the revised manuscript, GPC3^{KO} HepG2 cells and GPC3 expressing MCF7 cells were further tested. Please refer to the response there.

In addition to the supplied data, we would like to further clarify the reviewer's concern. Anti-GPC3 scFv can promote the interaction between eFT-CNVs and GPC3 overexpressing cells. The interaction between GPC3 and anti-GPC3 scFv allows eFT-CNVs to anchor onto the target cell membranes for a while. The anchorage allows fusogens have sufficient time to disassemble E1-E2 heterodimers and project E1 outwards. Subsequently, E1 can leisurely insert into target cell membranes and trigger membrane fusion. In comparison, cells with low level of GPC3 expression or null expression have much small chance of interaction or short detention time. Therefore, fusogens on eFT-CNVs may not be able to efficiently accomplish membrane fusion. For example, in blood vessel eFT-CNVs have small chance to attach onto vascular endothelial cells due to their poor expression of GPC3. Therefore, membrane fusion is unlikely to happen. However, in the following scenario membrane fusion may happen without the assistance of anti-GPC3 scFv. If eFT-CNVs were physically trapped on vascular endothelial cell membranes for a long time, the membrane fusion may finally happen.

On the other hand, normal tissues express a small amount of GPC3 to maintain normal cellular functions (*Nature Communications*, vol 6, 6536, 2015). Theoretically, eFT-CNVs may have on-target off-tumor effect. But the systemic toxicity is expected to be very low because the low expression of GPC3 in normal tissues. The expression level of GPC3 in normal tissue can be found here (https://www.cbioportal.org/results/plots?cancer_study_list=ccl_2019&Z_SCORE_THRESHOLD=2.0&RPPA_SCORE_THRESHOLD=2.0&profileFilter=mutations%2Cfusion%2Ccna&case_set_id=ccl_2019_cnaseq&gene_list=GPC3&geneset_list=%20&tab_index=tab_visualize&Action=Submit&plots_horz_selection=%7B%22selectedGeneOption%22%3A2719%2C%22dataType%22%3A%22clinical_attribute%22%2C%22selectedDataSourceOption%22%3A%22CANCER_TYPE_DETAILED%22%7D&plots_vert_selection=%7B%22selectedGeneOption%22%3A2719%2C%22dataType%22%3A%22MRNA_EXPRESSION%22%2C%22selectedDataSourceOption%22%3A%22rna_seq_mrna%22%2C%22logScale%22%3A%22true%22%7D&plots_coloring_selection=%7B%7D). The animal data can also demonstrate the specificity *in vivo*. We did not observe cytotoxicity in mouse major organs because mouse does not express human GPC3. Limited cytotoxicity was

observed in mouse liver simply because liver can nonspecifically uptake nanomaterials.

Membrane extrusion is an interesting strategy for scaling the production extracellular vesicle production. However, it seems possible that extruded membrane particles could be a mixture of particles, where some have the outer plasma membrane leaflet facing outward and some inward. Is there an experiment that the authors could do to get a sense of the proportion of particles that are displaying the targeting and fusogen molecules in the appropriate orientation?

Thanks for the reviewer's concern. We agree with the reviewer's comment. During membrane extrusion the disassembled lipids can spontaneously self-assemble and form CNVs, and CNVs could have plasma membrane leaflet facing inward. We fully admit this possibility. Getting back to the reviewer's question, it is very challenging to quantitatively detect the proportion of CNVs with outward-oriented targeting moieties and fusogens. Fluorescence-labeled antibody can detect the outer membrane-bound fusogens followed by measurement with a super-resolution flow cytometry. However, the existing flow cytometry cannot efficiently detect particles in 30-80 nm. Immunogold staining in combination with electron microscope and super-resolution microscope can detect the outer membrane-bound fusogens. However, the low-throughput analysis cannot provide reliably quantitative data. Actually, when the positively charged lipids and integral proteins on inner membranes flipped during mechanical extrusion and self-assembly of CNVs, these CNVs with positively charged surface can aggregate with CNVs with negatively charged surface. Therefore, the great majority of CNVs with plasma membrane leaflet facing inward could be removed during 16,500 g centrifugation and subsequent filtration. In the updated Supplementary Fig. 4b, we demonstrated the average zeta-potential of our nanovesicles was negative. The finding may indicate that the majority of CNVs still have plasma membrane leaflet facing outward. Moreover, in immunogold staining we observed gold nanoparticles on almost all CNVs. Altogether, it is difficult to detect the proportion of particles with flipped plasma membrane. But based on our speculation and the result of immunogold staining, we tend to think that the vast majority of harvested CNVs have the outward-oriented fusogens and targeting moieties.

More experimental detail should be provided on the membrane fragment extrusion process. The authors say that the supernatant was passed through 0.22- μ m pore filter. Please provide more information on the filter. Was a mini extruder used? A vacuum filter?

Thanks for the reviewer's concern. In the experimental section we described that "*Harvested cell pellets were resuspended in hypotonic buffer containing a proteinase phosphatase inhibitor cocktail followed by cell disruption with a Dounce homogenizer.*" In this study CNVs were generated by mechanical extrusion using a Dounce homogenizer. To collect CNVs with <200 nm in diameter for drug loading, a 0.22- μ m pore filter was further used. It was not a mini extruder. We did not use a vacuum filtration apparatus but processed the filtration manually. In the updated Supplementary Information, we added "*manually*" and supplied product information "*Sigma-Aldrich, SLGVR33RS*".

The authors should add a citation to line 288-289 describing the fusogen.

Thanks for the suggestion. We cited “Yang L, Bailey L, Baltimore D, Wang P. Targeting lentiviral vectors to specific cell types in vivo. *Proc Natl Acad Sci USA* 103, 11479-11484 (2006)” and “Morizono K, et al. Lentiviral vector retargeting to P-glycoprotein on metastatic melanoma through intravenous injection. *Nat Med* 11, 346-352 (2005)” in the updated manuscript.

The authors describe an experiment without showing the results (starting at 290). The data should be shown or the discussion of the result omitted.

Thanks for the suggestion. The recombinant fusion protein we constructed did not show distinct binding or fusion effect. Therefore, we did not show the negative results in the manuscript. Indeed, the design was too ambitious and naive without full understanding of the Sindbis virus fusion protein. The first two pictures below show the structure of wild-type Sindbis virus fusion protein. In this unsuccessful design, we directly replaced E2 extracellular domain-E3-capsid with anti-GPC3 scFv. The relevant design and AA sequence can be found below. HA tag was used for expression detection. The donor cells successfully overexpressed the chimeric proteins. However, these proteins did not show distinct binding or fusion activities. We speculated that GPC3 scFv may change the spatial conformation of E1 but also impair the binding capability of scFv. It looks like that E1-E2 heterodimer is essential, which can reserve E1 function. At acidic environment, the dissociation between E1 and E2 can trigger E1 homotrimerization and projection of E1 outwards. Subsequently, the fusion loops can be buried in the target cell membrane and initialized membrane fusion. Nevertheless, we provided the design information and relevant experimental data here for reviewers only. Do not really need to show the naïve design and nonspecifically negative results in this manuscript. In the manuscript we gave a brief discussion. The main purpose was to let the readers know that we tried with fusion proteins but failed to make it happen. If readers have the relevant protein engineering experience, they probably can construct the proposed fusion proteins.

KOZAK-SP(IgG kappa)-scFv-HA tag-partial E2 domain-6K-E1 domain

KOZAKMETDTLLLWVLLLWVPGSTGDDVVMQTPLSLPVSLGDQASISCRSSQSLVHSNGNTYLHWYLQKPG
 QSPKLLIYKVSNRFSQVDFRFSQSGSGTDFLLKISRVEAEDLGVYFCSQNTHPPTFGSGTKLEIKGGGSGG
 GGSGGGGSQVQLQQSGAELVRPGASVKLSCKASGYTFTDYEMHWVKQTPVHGLKWIGALDPKTGDTAYSQ
 KFKGKATLTADKSSSTAYMELRSLTSEDSAVYYCTRFYSYTYWGGQGLTVTSAMYPYDVPDYAVYITLAVASA
 TVAMMIGVTVAVLCACKARRECLTPYALAPNAVPTSLALLCCVRSANAETFTETMSYLWSNSQPFVWVQLCIP
 LAAFIVLMRCCSCCLPFLVAGAYLAKVDAIEHATTVPNVPQIPYKALVERAGYAPLNLEITVMSSEVLPSTNQE
 YITCKFTTVVPSPIKCCGSLECPAAHADYTCVKVFGGVYPMWGGACFCDSSENSQMSEAYVELSADCASD
 HAQAIKVHTAAMKVGRLRIVYGNNTSFLDVYVNGVTPGTSKDLKVIAGPISASFTPFDHKVVIIHRGLVYNYDFPEY
 GAMKPGAFGDIQATSLTSKDLIASTDIRLLKPSAKNVHVPYDQASSGFEMWKNNSGRPLQETAPFGCKIAVNPL

RAVDCSYGNIPISIDIPNAAFIRTSAPLVSTVKCEVSECTYSADFGGMATLQYVSDREGQCPVHSHSSTATLQ
ESTVHVLEKGAVTVFHSTASPQANFIVSLCGKKTTCNAECKPPADHIVSTPHKNDQEFQAAISKTSWSWLFALF
GGASSLLIIGLMIFACSMMLTSTRR

① HEK293-SP+SP(IgG kappa)-scFv-HA tag-partial E2 domain-6K-E1 melt curve

③ HEK293-CTRL+SP(IgG kappa)-scFv-HA tag-partial E2 domain-6K-E1 melt curve

Line 81/82 of the methods needs a citation.

Thanks for the suggestion. We cited “Kim MS, et al. Development of exosome-encapsulated paclitaxel to overcome MDR in cancer cells. *Nanomedicine* 12, 655-664 (2016)” and “Wan Y, et al. Aptamer-Conjugated Extracellular Nanovesicles for Targeted Drug Delivery. *Cancer Res* 78, 798-808 (2018)” in the updated Supplementary Information.

REVIEWERS' COMMENTS

Reviewer #1 (Remarks to the Author):

The authors have addressed all my concerns. Nice work.

Reviewer #2 (Remarks to the Author):

The author has answered the reviewer's questions and suggestions satisfactorily experimentally and logically, and the reviewer recommends approval for publication in Nature Communications.

Reviewer #3 (Remarks to the Author):

The revised manuscript is significantly improved. A crucial concern regarding the approach was whether the antibody fragment was delivering specifically to cells that express the cognate ligand, or if the delivery was independent of scFv binding. Two reviewers suggested that it would have been informative if the authors had tested CNV displaying a fusion molecule lacking a targeting molecule (or displaying a control scFv plus fusogen). However, in the updated manuscript, the authors have presented new data demonstrating scFv-targeted particles effectively deliver to HepG2 cells (which express GPC3) but not to GPC3-KO HepG2 cells. This finding is compelling and provides convincing evidence for the importance of the scFv in in vitro experiments.